# Control of craniofacial development by the collagen receptor, discoidin domain receptor 2

Fatma F Mohamed[1], Chunxi Ge[1†], Shawn A Hallett[1†], Alec C Bancroft[2], Randy T Cowling[3], Noriaki Ono[4], Abdul-Aziz Binrayes[5], Barry Greenberg[3], Benjamin Levi[2], Vesa M Kaartinen[6], Renny T Franceschi[1,7,8*]

[1]Department of Periodontics & Oral Medicine, University of Michigan School of Dentistry, Ann Arbor, United States; [2]Center for Organogenesis and Trauma, Department of Surgery, University of Texas Southwestern, Dallas, United States; [3]Division of Cardiovascular Medicine, University of California, San Diego, San Diego, United States; [4]Department of Diagnostic and Biomedical Sciences, University of Texas Health Science Center at Houston School of Dentistry, Houston, United States; [5]Department of Prosthetic Dental Sciences, College of Dentistry, King Saud University, Riyadh, Saudi Arabia; [6]Department of Biologic & Materials Science, University of Michigan School of Dentistry, Ann Arbor, United States; [7]Department of Biological Chemistry, School of Medicine, University of Michigan, Ann Arbor, United States; [8]Department of Biomedical Engineering, University of Michigan, Ann Arbor, United States

*For correspondence:
rennyf@umich.edu

†These authors contributed equally to this work

**Abstract** Development of the craniofacial skeleton requires interactions between progenitor cells and the collagen-rich extracellular matrix (ECM). The mediators of these interactions are not well-defined. Mutations in the discoidin domain receptor 2 gene (*DDR2*), which encodes a non-integrin collagen receptor, are associated with human craniofacial abnormalities, such as midface hypoplasia and open fontanels. However, the exact role of this gene in craniofacial morphogenesis is not known. As will be shown, *Ddr2*-deficient mice exhibit defects in craniofacial bones including impaired calvarial growth and frontal suture formation, cranial base hypoplasia due to aberrant chondrogenesis and delayed ossification at growth plate synchondroses. These defects were associated with abnormal collagen fibril organization, chondrocyte proliferation and polarization. As established by localization and lineage-tracing studies, *Ddr2* is expressed in progenitor cell-enriched craniofacial regions including sutures and synchondrosis resting zone cartilage, overlapping with GLI1 + cells, and contributing to chondrogenic and osteogenic lineages during skull growth. Tissue-specific knockouts further established the requirement for *Ddr2* in GLI +skeletal progenitors and chondrocytes. These studies establish a cellular basis for regulation of craniofacial morphogenesis by this understudied collagen receptor and suggest that DDR2 is necessary for proper collagen organization, chondrocyte proliferation, and orientation.

## Editor's evaluation

This fundamental work substantially advances our understanding of the development of the craniofacial skeleton requires interactions between progenitor cells and the collagen-rich extracellular matrix (ECM). The evidence supporting the conclusions is compelling, with rigorous biochemical assays, state-of-the-art localization and lineage tracing studies, and phenotype and genetic analysis of Ddr2-deficient mice. The work will be of broad interest to cell biologists and developmental biologists.

## Introduction

Craniofacial abnormalities affecting bone formation in the skull and face are common birth defects in humans (*Fish, 2016*; *Trainor and Richtsmeier, 2015*). The craniofacial skeleton forms through regulated bone growth in the calvarium synchronized with growth of the developing brain and cranial base. Cranial sutures, fibrous tissues adjoining calvarial bones, are the principal sites for growth of the cranial vault. Sutures contain stem cells associated with several genetic markers including glioma-associated oncogene 1 (*Gli1*), an intermediate in hedgehog signaling, Axin-related protein 2 (*Axin2*), a component of the Wnt pathway, and paired related homeobox 1 (*Prrx1*) (*Lana-Elola et al., 2007*; *Maruyama et al., 2016*; *Wilk et al., 2017*; *Zhao et al., 2015*). The current paradigm suggests that sutural stem cells allow continued calvarial growth through osteogenic differentiation and bone formation at sutural edges of the calvarial bones. This process is coordinated with growth of the cranial base mediated by endochondral ossification centers called synchondroses. Histologically, the synchondrosis is a mirror-image growth plate with resting chondrocytes in the central zone and proliferative and hypertrophic chondrocytes on both sides. During endochondral ossification, the resting chondrocytes in the growth plate synchondrosis undergo sequential proliferation, hypertrophic differentiation, and apoptosis before being replaced by bone. The cranial base has been implicated in many human syndromes such as craniosynostosis (premature closure of cranial sutures), Down syndrome, Turner syndrome, cleidocranial dysplasia, cleft palate, and osteogenesis imperfecta (*Cheung et al., 2011*; *McGrath et al., 2012*; *Nie, 2005*; *Paliga et al., 2014*; *Tahiri et al., 2014*).

The extracellular matrix (ECM) is critical for maintenance of stem cell niches, differentiation, organ development, and remodeling (*Lu et al., 2011*). During craniofacial development, skeletal stem cells interact with the collagen-rich ECM through cell-surface receptors. These interactions regulate cell proliferation, migration, differentiation, and remodeling to control and maintain shapes of individual cranial bones and the relative proportions of skeletal elements (*Erlebacher et al., 1995*). Disruption of these interactions can cause craniofacial abnormalities (*Lu et al., 2011*; *Erlebacher et al., 1995*; *Velleman, 2000*; *Kuivaniemi et al., 1997*). Discoidin domain receptors (DDR1 and DDR2) are a unique class of receptor tyrosine kinases (RTKs) that, together with β1 integrins, mediate interactions between cells and the ECM by specifically binding triple-helical collagens (*Vogel et al., 1997*; *Shrivastava et al., 1997*; *Leitinger and Hohenester, 2007*). DDRs contain a unique discoidin (DS) domain on their extracellular surface that is homologous to discoidin I-like domain of the slime mold *Dictyostelium discoideum* (*Leitinger and Hohenester, 2007*). DS domains are required for collagen recognition and specificity of DDRs such that DDR2 preferentially binds to fibrillar collagens I-III, but not nonfibrillar basement membrane type IV collagen that is exclusively recognized by DDR1 (*Vogel et al., 1997*; *Shrivastava et al., 1997*; *Leitinger and Kwan, 2006*).

*DDR2* plays an important role during human craniofacial morphogenesis, possibly being involved in skull changes occurring during the evolution of modern humans from archaic hominins (*Kuhlwilm and Boeckx, 2019*). Moreover, *DDR2* loss of function mutations cause the autosomal recessive growth disorder, spondylo-meta-epiphyseal dysplasia, with short limbs and abnormal calcifications (SMED, SL-AC). This disorder is associated with a number of craniofacial abnormalities including a prominent forehead, open fontanelles, hypertelorism, a short nose with a depressed nasal bridge, long philtrum and micrognathia (*Alkindi et al., 2014*; *Ali et al., 2010*; *Bargal et al., 2009*; *Mansouri et al., 2016*; *Ürel-Demir et al., 2018*; *Smithson et al., 2009*; *Rozovsky et al., 2011*). *Ddr2*-deficient mice recapitulate craniofacial phenotypes seen in SMED SL-AC patients including short snout, protruding eyes, abnormal tooth development and impaired calvarial ossification (*Ge et al., 2016*; *Kano et al., 2008*; *Labrador et al., 2001*; *Mohamed et al., 2020*). However, the basis for these abnormalities is not understood.

We recently described cell autonomous functions of *Ddr2* in growth of the appendicular skeleton (*Mohamed et al., 2022*). Preferential localization of DDR2 in GLI1-positive skeletal progenitor populations associated with resting/growth zone chondrocytes of the growth plate and bone marrow was demonstrated while conditional inactivation of *Ddr2* in *Gli1*-expressing cells and chondrocytes phenocopied the dwarf bone phenotype of globally *Ddr2*-deficient mice. In the present study, we focus on DDR2 functions in craniofacial development. We show that DDR2 is associated with putative suture stem cells that can serve as precursors for osteoblasts/osteocytes in the cranial vault and with synchondrosis-associated resting/proliferating chondrocytes that form hypertrophic chondrocytes, osteoblasts, and osteocytes in the cranial base. Furthermore, as shown by global and conditional

**eLife digest** We each have unique facial features that are key to our identities. These features are inherited, but the mechanisms are poorly understood. People with the genetic disease spondylo-meta-epiphyseal dysplasia, or SMED, have characteristic facial and skull abnormalities including a flattened face and shortened skull. SMED is associated with mutations that inactivate the gene encoding a protein called discoidin domain receptor 2 (DDR2), which is a receptor for collagen.

Collagen is the major structural protein in the human body, supporting the structure of cells and tissues. It also controls cell behaviors including growth, migration and differentiation, and it helps form tissues such as cartilage or bone. At least some of the effects of collagen on cells depend on its interaction with DDR2.

Since the facial and skull abnormalities in mice with mutations that stop DDR2 from working correctly resemble those of SMED patients, these mice can be used to understand the cellular basis for this disease, as well as the role of DDR2 in the embryonic development of the face and skull. Therefore, Mohamed et al. set out to understand how loss of DDR2 causes the characteristic facial and skull defects associated with SMED.

Mohamed et al. used mice that had been genetically modified so that DDR2 could be inactivated in skeletal progenitor cells, cartilage cells and bone cells (osteoblasts). Examining these mice, they found that the shortened skulls and flat face characteristic of mice lacking DDR2 are due to bones at the skull base failing to elongate correctly due to defects in the growth centers that depend on cartilage. Mohamed et al. also discovered that the cells that normally produce DDR2 are the progenitors of cartilage and bone-forming cells, which partly explains why lacking this protein leads to issues in growth of these tissues.

In addition to shedding light on the causes of SMED, Mohamed et al.'s results also provide general insights into the mechanisms controlling the formation of facial and skull bones that depend on interactions between cells and collagen. This information may help explain how other abnormalities in the face and skull emerge, and provide a basis for how the shape of the skull has changed during human evolution. In the future, it may be possible to manipulate the activity of DDR2 to correct skull defects.

inactivation approaches, DDR2 controls suture formation, frontal bone thickness and anterior/posterior growth of the skull. Notably, these DDR2 functions involve activity in different cell populations and are correlated with changes in collagen organization and chondrocyte polarity.

## Results

### Anterior-posterior skull growth and frontal suture/bone formation impairment in *Ddr2*-deficient mice

To begin understanding the function of *Ddr2* in craniofacial development, we first defined which parts of skull are altered in global *Ddr2* deficiency using *Ddr2*[slie/slie] mice. These animals contain a spontaneous 150 kb deletion in the *Ddr2* locus to generate an effective null (*Kano et al., 2008*). We performed linear measurements on micro-CT scans of 3-month-old skulls (n=10) using previously described landmarks (*Vora et al., 2015*). Our analysis along the anterior-posterior (AP) axis revealed a significant reduction (12%) in skull length (SL) mainly due to shortening of the nasal bone (NB), cranial vault (CV), and anterior and posterior cranial base (ACB and PCB) in *Ddr2*-deficient mice compared with WT littermates (*Figure 1a–c*). In addition, we detected an increase in anterior skull width without any change in skull height (*Figure 1d–e*). We further selected orthogonal planes in MicroView to measure the thickness of calvarial bones (*Figure 1f*). Interestingly, frontal bone thickness was reduced by 55% with *Ddr2* deficiency; however, no significant differences were observed in the thickness of parietal or occipital bones (*Figure 1f–g*). Further analysis using Alcian blue and Alizarin red whole mount and H&E staining of 2-week-old skulls showed defective formation of frontal sutures in *Ddr2*-deficient mice (defect in 3/3 mice examined). In contrast, coronal and lambdoid sutures had a normal morphology (*Figure 1h–i*). *Ddr2*-deficient calvaria also had a reduced calvarial bone marrow cavity that was most prominent in the anterior skull bones (*Figure 1f and i*). No major differences were observed in transverse cranial sutures, such as coronal and lambdoid. Consistent with results from

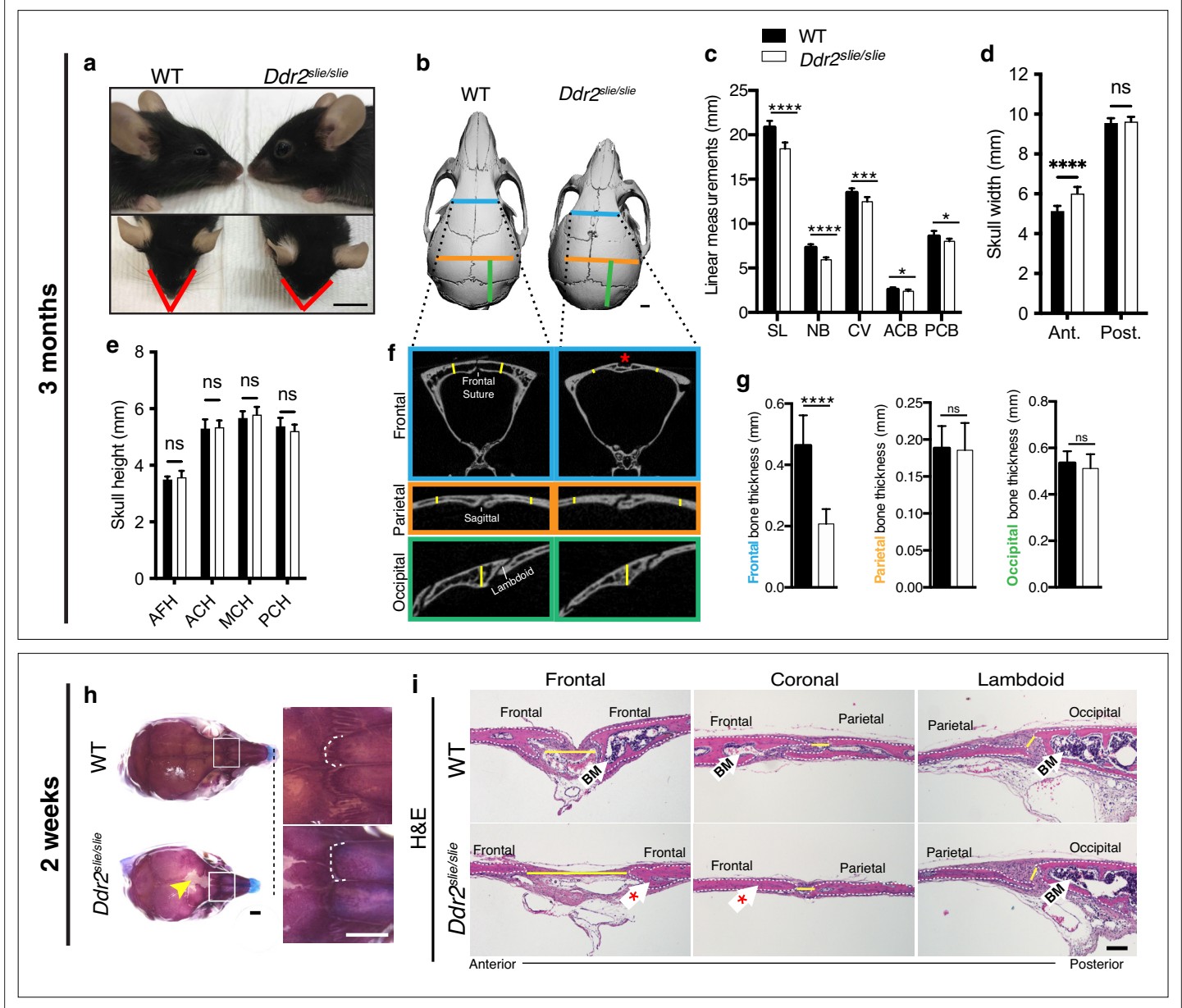

**Figure 1.** *Ddr2* deficiency results in impaired anterior-posterior skull growth with abnormal frontal bone and suture formation. WT and *Ddr2slie/slie* mice were compared at 3 months (**a–g**) and 2 weeks (**h,i**). (**a-c**) Short snout, and reduced skull length in Ddr2slie/slie mice. (**a**) Side (upper) and top (lower) head views of 3-month-old *Ddr2slie/slie* mice and WT littermates. Scale bar: 1 cm. (**b**) 3D rendering of μCT scans of 3-month-old skulls. Scale bar: 1 mm. (**c**) Linear measurements along anteroposterior axis of skulls, where SL: skull length; NB: nasal bone; CV: calvaria vault; ACB: anterior cranial base; PCB: posterior cranial base. (**d**) Quantification of anterior (ant.) and posterior (post.) skull width showed a selective increase only in the anterior skull of *Ddr2slie/slie* vs WT mice. (**e**) No changes were observed in skull height at any of the regions measured (anterior cranial height, ACH; middle cranial height, MCH; posterior cranial height, PCH). (**f,g**) μCT scans of calvarial bones and quantification showing a significant reduction of frontal bone (blue) in 3-month-old *Ddr2slie/slie* mice in the absence of changes in parietal (orange) or occipital (green) calvarial bones. Note frontal suture defect in *Ddr2slie/slie* mice (red asterisk). Data are presented as mean ± SD. (n=10). *p<0.01, **p<0.01, ***p<0.001, ****p<0.0001, ns, not significant, two-tailed unpaired *t* test. (**h**) Alcian blue and Alizarin red staining of 2-week-old mouse skull whole mounts shows delayed frontal suture formation (yellow arrow) and abnormal suture morphology (white dotted lines) in *Ddr2slie/slie* mice compared with WT. Boxed region is shown in higher magnification; right. Scale bar: 100 μm. (**i**) Hematoxylin and eosin (**H&E**) staining shows open frontal sutures in *Ddr2slie/slie*, but transverse sutures, such as coronal and lambdoid were not affected (highlighted by yellow lines). *Ddr2slie/slie* calvariae also had a smaller bone marrow cavity (Bm, white arrows) compared with WT. Frontal suture, coronal section; transverse sutures, sagittal section. Scale bar: 100 μm.

The online version of this article includes the following source data and figure supplement(s) for figure 1:

**Source data 1.** Skull linear measurements in Wildtype versus *Ddr2* knockout mice.

**Figure supplement 1.** Orientation of skulls in sagittal, axial, and coronal planes.

2-week-old mice, reduced mineralization in the posterior frontal suture region was also seen in most skulls from 3-month-old *Ddr2*[slie/slie] mice, although some phenotypic variability was observed at this age with 6/10 skulls affected (*Figure 1b*). In summary, skull deformities in *Ddr2*-deficient mice are associated with defects in AP skull growth, impaired frontal suture mineralization, and frontal bone thickness.

## Reduced anterior-posterior skull growth in *Ddr2* deficiency is explained by abnormal chondrogenesis and delayed synchondrosis ossification

Skull elongation is mainly driven by endochondral ossification at the cranial base (*Hallett et al., 2022*). The impaired AP skull growth observed in *Ddr2*[slie/slie] mice prompt us to examine the cranial base synchondroses and associated bones. We focused on intersphenoid (ISS) and spheno-occipital synchondroses (SOS) that form the anterior and posterior borders of basisphenoid bone (BS), the most affected cranial base bone in *Ddr2*[slie/slie] mice (*Figure 2c*). The BS bone forms the mid-posterior cranial base providing a foundation to support the brain, along with presphenoid (PS) and basio-occipital (BO) bones (*Figure 2a*). Our analysis showed that the ISS and SOS were abnormally wide in *Ddr2*[slie/slie] mice compared with WT littermates (*Figure 2a–d*). At 2 months, the width of the ISS and SOS was increased by 187% and 23%; respectively, while a significant increase in the height was only observed in the SOS of knockout mice (12%) (*Figure 2b*). Accordingly, growth of basisphenoid bone was significantly inhibited in the knockout mice (reduced by 14%) (*Figure 2c*). The basio-occipital (BO) bone was also significantly affected but to lesser extent. No major difference was seen in the presphenoid (PS) bone (*Figure 2c*). H&E staining of 2-week-old skulls revealed that the *Ddr2*-deficient synchondroses, particularly ISS, showed severe defects in chondrocyte organization associated with loss of the columnar arrangement of proliferative chondrocytes (*Figure 2d*).

To begin understanding the cellular basis for the observed changes in the size of cranial base bones and synchondrosis structure, we assessed cell proliferation and apoptosis. EdU staining (shown in green) indicated labeling mainly in the proliferative chondrocyte zone of ISS and SOS (*Figure 2e*, white arrowheads); however, EdU +chondrocytes were significantly reduced in *Ddr2*-deficient synchondroses: 55% in ISS and 26% in SOS, compared with WT controls (*Figure 2f*). We also performed TUNEL assay to measure apoptosis, but did not detect differences between *Ddr2*-deficient and control mice in either ISS or SOS regions (*Figure 2g–h*). Consistent with this result, staining for cleaved caspase 3 was also not affected by *Ddr2* deficiency (*Figure 2—figure supplement 1*).

## *Ddr2* deficiency is associated with abnormal cranial base ECM organization

While a reduction in chondrocyte proliferation may explain the smaller BS and BO bones seen in *Ddr2*-deficient mice, proliferation changes do not readily explain the observed synchondrosis widening. To gain further insight into the pathogenesis underlying this phenomenon, we examined the distribution of ECM proteins associated with resting and proliferative chondrocytes (type II collagen), hypertrophic chondrocytes (type X collagen) and osteoblasts (type I collagen and integrin-binding sialoprotein (IBSP)) (*Robey, 2002*).

In wild-type mice, immunofluorescent staining of type II collagen was homogenously distributed in the cartilage matrix around chondrocytes. However, Col II was unevenly distributed in *Ddr2*-deficient synchondroses where more intense territorial matrix staining was observed around small clusters of chondrocytes with diminished staining in the interterritorial matrix (*Figure 3a*, **arrows**). These data suggest that DDR2 may regulate Col II fibril distribution and/or orientation. To explore this further, second harmonic generation (SHG) imaging was used to determine collagen fibril orientation in the central resting zone of ISS from WT and *Ddr2*[slie/slie] mice (*Figure 3—figure supplement 1*). In control synchondroses, collagen fibrils were evenly distributed and generally oriented along the A-P axis of the ISS. In contrast, fibrils in *Ddr2*-deficient synchondroses were more concentrated adjacent to cells and had a more randomized orientation as reflected by a dramatic decrease in anisotropy relative to WT controls.

Type X collagen immunostaining showed a specific signal in the hypertrophic zone although no consistent differences in staining intensity were observed between WT and *Ddr2*[slie/slie] mice (*Figure 3b*). Immunofluorescence using COL I and IBSP antibodies showed specific staining in trabecular (Tb, primary spongiosa) and cortical (Ct) bones (*Figure 3cd*, *Figure 3—figure supplement 2*

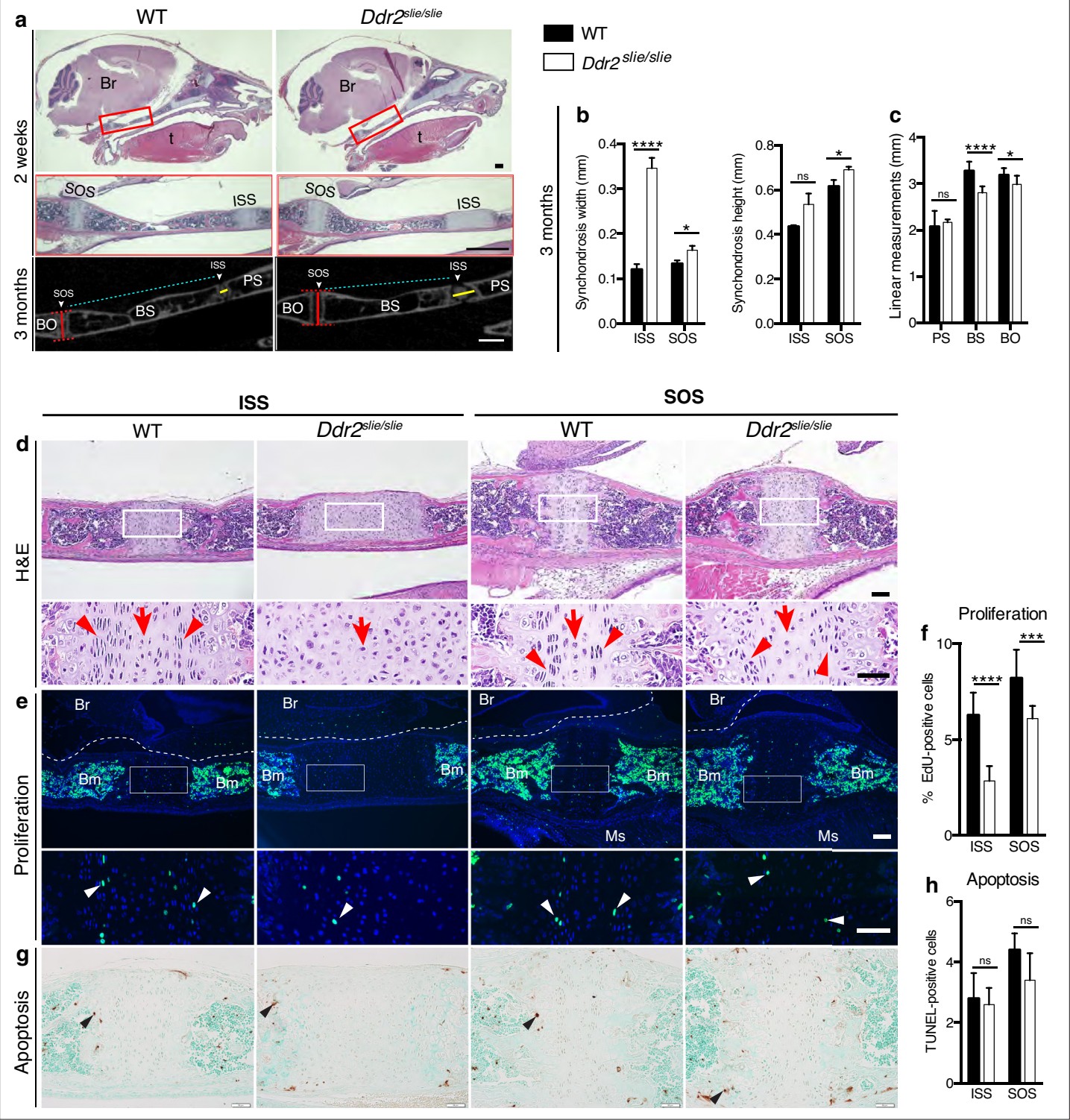

**Figure 2.** Cranial base hypoplasia due to chondrocyte disorganization and reduced chondrocyte proliferation in *Ddr2*-knockout synchondroses. (**a**) H&E staining (upper) and μCT scans (lower) of WT and *Ddr2slie/slie* skulls showing wide cranial base synchondroses. Scale bar: 500 μm. Boxed region (red) is shown in higher magnification. Br: Brain; t: Tongue. ISS: Interspheniod synchondrosis; SOS: Spheno-occipital synchondrosis; PS: Presphenoid bone; BS: Basisphenoid bone; BO: Basis-occipital bone. In μCT scan of skulls (lower), arrowheads point to cranial base synchondroses; yellow and red lines highlight the width and height of synchondroses; respectively (Quantification is shown in **b**). Cyan lines highlight shortening of basisphenoid bone between ISS and SOS in *Ddr2slie/slie* vs WT mice (Quantification of cranial base bone lengths is shown in **c**); Data are presented as mean ± SD. (n=10). *p<0.01, ****p<0.0001, ns, not significant, two-tailed unpaired *t* test. (**d**), H&E staining of ISS and SOS sections showing loss of columnar organization of

*Figure 2 continued on next page*

*Figure 2 continued*

proliferative chondrocytes (red arrowheads) in *Ddr2*<sup>slie/slie</sup> mice at 2 weeks of age. Red arrows point to resting chondrocyte zone. Boxed region is shown in higher magnification. Scale bar: 100 µm. (**e**) EdU staining (green) of ISS and SOS sections showing reduction in EdU + cells (white arrowheads) in *Ddr2*<sup>slie/slie</sup> mice compared with WT littermates. Boxed region is shown in higher magnification. Scale bar: 100 µm. Br: Brain, Ms: Muscle; Bm: Bone marrow. White dotted lines define the ventral surface of brain. (**f**) Percentage of EdU + cells in ISS and SOS of WT and *Ddr2*<sup>slie/slie</sup> mice. (**g**) TUNEL staining (brown, black arrowheads) shows no changes in apoptotic levels between mice *Ddr2*<sup>slie/slie</sup> and WT. Cell nuclei were stained with methyl green (green). Scale bar: 50 µm. (**h**) Quantification of TUNEL-positive cells in cranial base synchondroses. Data in are presented as mean ± SD (panel f, n=5-7; h, n=5). ***$p<0.001$, ****$p<0.0001$, ns, not significant, two-tailed unpaired $t$ test.

The online version of this article includes the following source data and figure supplement(s) for figure 2:

**Source data 1.** Cranial base linear measurements in Wildtype versus *Ddr2* knockout mice.

**Source data 2.** Proliferation (EdU labeling) and apoptosis (TUNEL assay) in Wildtype versus *Ddr2* knockout mice.

**Figure supplement 1.** No change in levels of cleaved caspase 3 levels in wild-type versus *Ddr2*<sup>slie/slie</sup> mice.

for quantification of staining intensity). For COL I, no differences in staining intensity were observed when WT and *Ddr2*<sup>slie/slie</sup> mice were compared. However, IBSP staining was reduced in hypertrophic chondrocytes and primary spongiosa in *Ddr2*-deficient SOS compared with WT, suggesting defective bone formation and mineralization, although these differences were not seen in the ISS. In summary, *Ddr2* deficiency is associated with abnormal chondrogenesis in the cranial base, disorganized chondrocytes, reduced chondrocyte proliferation, abnormal COL2 ECM distribution, randomized fibril orientation and delayed endochondral ossification.

## Distribution of *Ddr2* expression in the craniofacial skeleton

To begin relating the cellular functions of DDR2 to the observed craniofacial phenotype of *Ddr2*-deficient mice, we first examined the temporal and spatial distribution of *Ddr2*-expressing cells using a *Ddr2*-lacZ knock-in mouse model. The expression pattern of *Ddr2* was examined using a combination of whole mount and frozen sections from *Ddr2*<sup>+/LacZ</sup> mice. During fetal development, LacZ activity was first detected at E11.5 (not seen at E9.5 -not shown). The whole mount staining at this time revealed broad *Ddr2* expression in the developing midface including the median and lateral nasal process (MNP and LNP), maxillary and mandibular processes, and around eyes (*Figure 4a*). A similar expression pattern was seen at E13.5 and E16.5 (*Figure 4a*). Using frozen sections of the E13.5 embryonic head, LacZ activity was detected in cartilage primordia of the cranial base, tongue mesenchyme, nasal septum, and Meckel's cartilage in the developing mandible, and, as previously reported, developing tooth buds (*Mohamed et al., 2020*). However, no LacZ activity was detected in the developing brain (*Figure 4—figure supplement 1a*).

Whole mounts of *Ddr2*<sup>+/LacZ</sup> skulls from newborn mice showed broad *Ddr2* expression across the calvarial surface with preferential enrichment in cranial sutures. This distribution became more restricted to sutures in 3-month-old adult mice (*Figure 4a*). Frozen sections of neonatal *Ddr2*<sup>+/LacZ</sup> skulls revealed *Ddr2* expression in the suture mesenchyme, periosteum of flanking bones, and dura mater on the ventral surface of calvaria (*Figure 4b*). Suture and periosteal expression were also seen in adult calvaria as well as in cells lining the bone marrow (*Figure 4—figure supplement 1b*). However, LacZ activity was not seen in osteocytes, the terminally differentiated cells inside calvarial bones. This indicates that *Ddr2* expression is highest during early stages of osteoblast differentiation. *Ddr2* expression was also detected in synchondroses, primarily located in the resting (RZ) and proliferative chondrocyte (PZ) zone but low or undetected in terminal hypertrophic chondrocytes (HZ), and in the associated bone marrow and periosteum (*Figure 4c*). Overall, *Ddr2* expression was highest in regions enriched in skeletal progenitor cells including cranial sutures, periosteum and dura mater and resting chondrocytes. This suggests that *Ddr2* has functions in skeletal progenitor cells which contribute to development of the craniofacial skeleton.

## DDR2-expressing cells colocalize with GLI1 in cranial sutures and synchondroses, contributing to osteogenic and chondrogenic lineages

Cranial sutures contain stem cells that contribute to craniofacial bone formation (*Zhao et al., 2015*). Since *Ddr2* is also expressed in sutures, we determined whether DDR2 is in skeletal progenitor cells whose progeny can form the major cranial bone cell types. To answer this question, we used

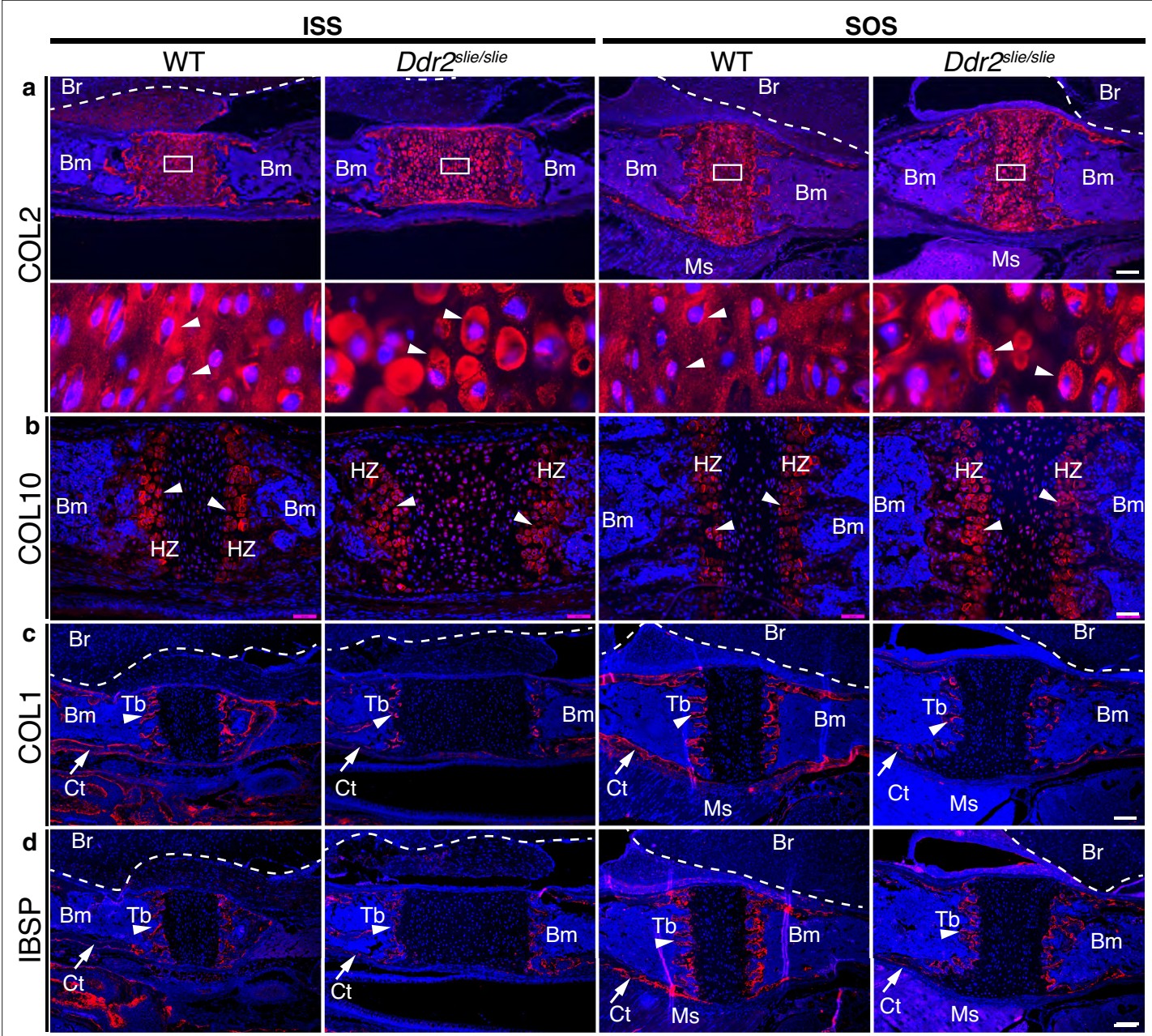

**Figure 3.** ECM defects in *Ddr2*-deficient synchondroses. (**a-d**) Immunofluorescent staining of ISS and SOS sections from 2-week-old WT and *Ddr2*<sup>slie/slie</sup> synchondroses. (**a**) Representative images of COL2 immunostaining (red) in ISS and SOS showing homogenous distribution around chondrocytes in WT synchondroses, while *Ddr2*<sup>slie/slie</sup> mice showed uneven, ring-like immunostaining around chondrocytes (white arrowheads). Boxed region is shown in higher magnification (bottom). Scale bar: 200 µm. (**b**) Immunofluorescent images of COL10 immunostaining in the hypertrophic zone (HZ) of synchondroses showing no major changes in staining distribution between WT and *Ddr2*<sup>slie/slie</sup> mice. Scale bar: 50 µm. (**c**) Immunofluorescence images showing COL1 (red) staining in trabecular (arrowheads) and cortical (arrows) bones of the cranial base in WT and *Ddr2*<sup>slie/slie</sup> mice. Scale bar: 200 µm. (**d**) Immunofluorescent images showing IBSP (red) in trabecular (arrowheads) and cortical (arrows) bones of cranial base is decreased in *Ddr2*<sup>slie/slie</sup> synchondrosis compared with WT littermates. See *Figure 3—figure supplement 2* for quantitation of IF staining. Scale bar: 200 µm. Cell nuclei were stained with DAPI (blue) in **a–d**. Bm: Bone marrow; Br: Brain; Tb: Trabecular bone; Ct: Cortical bone; Ms: Muscle.

The online version of this article includes the following figure supplement(s) for figure 3:

**Figure supplement 1.** Ddr2 deficiency is associated with loss of collagen organization in the ISS of *Ddr2*<sup>slie/slie</sup> mice.

**Figure supplement 2.** Quantitation of Ibsp (Bsp) and Col I immunofluorescence in WT and *Ddr2*<sup>slie/slie</sup> mice.

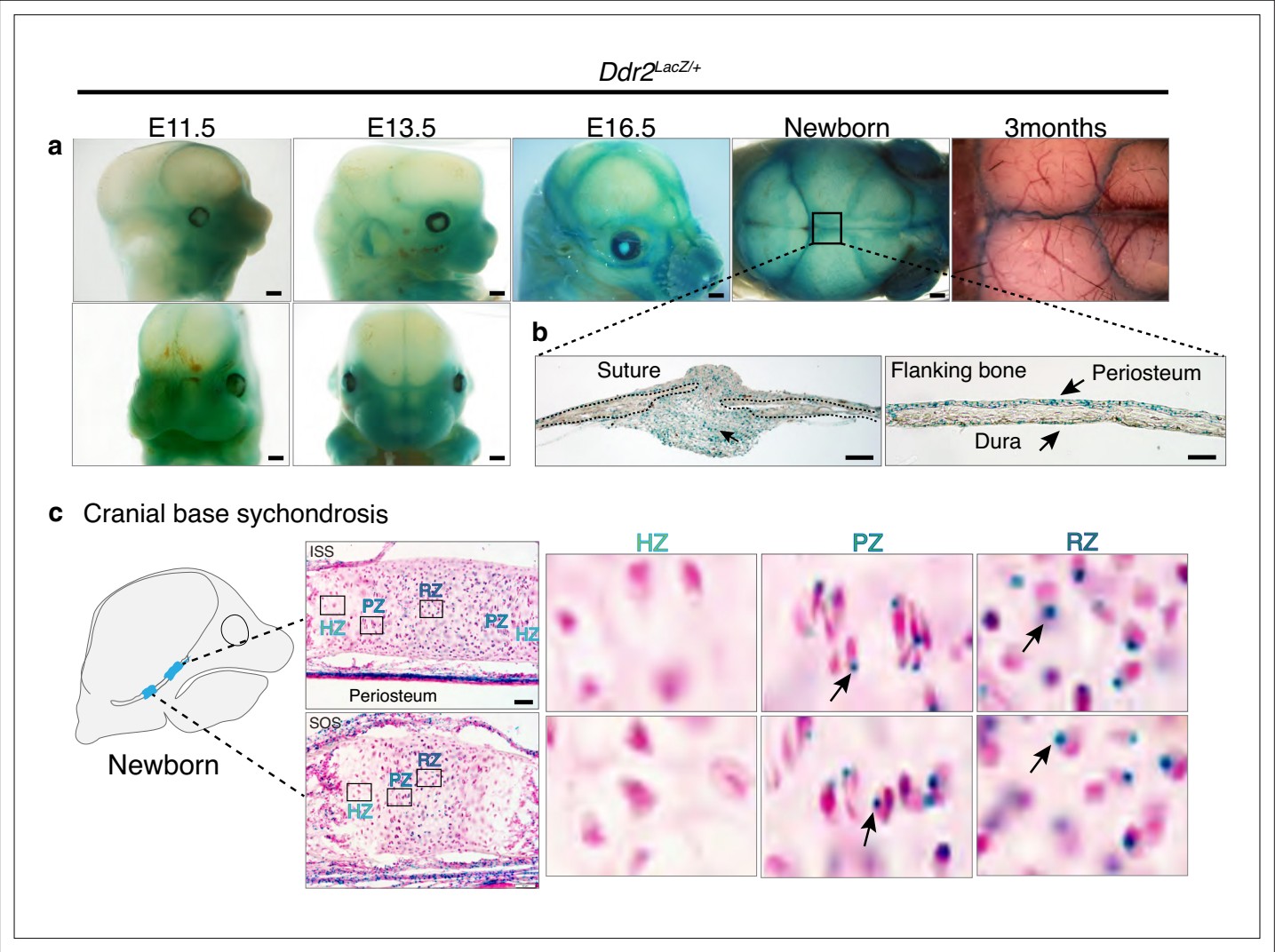

**Figure 4.** *Ddr2* expression in craniofacial skeleton. (**a**) Whole-mount X-gal staining (green) of *Ddr2*<sup>Lacz/+</sup> skulls showing of *Ddr2* expression in midfacial region, cranial vault, and cranial sutures. Scale bar: 50 μm. (**b**) X-gal staining of cryostat sections of calvaria from newborn mice showing expression in suture mesenchyme, periosteum, and dura mater of flanking bones. Scale bar: 100 μm, left and 50 μm, right. (**c**) X-gal staining of cryostat section of ISS (top) and SOS (bottom) from newborn mice revealing *Ddr2* expression in resting and proliferative chondrocyte zones, but low or undetected in terminal hypertrophic chondrocytes. Boxed regions are shown in higher magnification, right. Scale bar: 50 μm.

The online version of this article includes the following figure supplement(s) for figure 4:

**Figure supplement 1.** Ddr2-LacZ localization in craniofacial skeleton.

a lineage-tracing approach by breeding heterozygous *Ddr2*<sup>Mer-icre-Mer</sup> female mice with male homozygous *Rosa26*<sup>LSL-tdTomato</sup> reporter mice. Cre-mediated recombination in *Ddr2*<sup>Mer-icre-Mer</sup>;*Rosa26*<sup>LSL-tdTomato</sup> mice was induced with intragastric tamoxifen injections given for 4 days after birth and then skulls were harvested and analyzed at day 5, 14, and 2 months of age (***Figure 5a***). Whole mounts of calvaria showed labeling that persisted for at least 2 months in all cranial sutures; frontal, sagittal, coronal and occipital sutures (***Figure 5—figure supplement 1a***). At day 5, frozen sections revealed tdTomato labeling in a few cells in the cranial suture mesenchyme and in developing periosteum and dura mater of flanking calvarial bones (***Figure 5b***). At this age, the calvarial bone is very thin and devoid of a marrow cavity. Two weeks later, labeling was evident in cranial sutures, periosteum, and dura mater. Over a 2-month chase period, labeling became intense in cranial suture mesenchyme, where undifferentiated cells reside, in the lining cells in the bone marrow of flanking calvarial bones, and in osteocytes (***Figure 5b***, ***Figure 5—figure supplement 1a***). Therefore, the osteoblasts, osteocytes, and bone marrow-lining cells responsible for calvarial bone formation are derived from *Ddr2*-expressing cells.

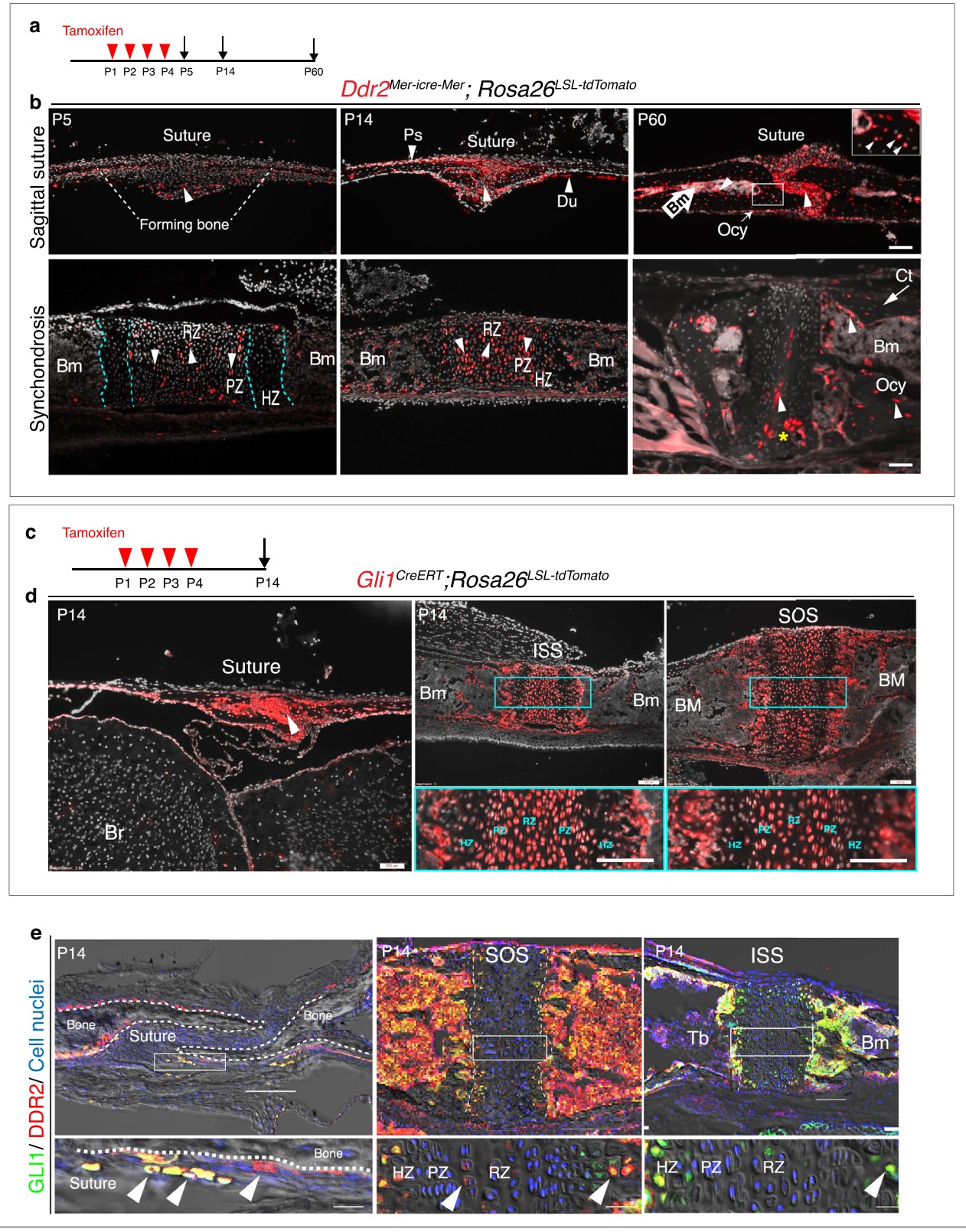

**Figure 5.** *Ddr2^{mer-iCre-mer}* marks progenitors of the skeletal lineage during postnatal craniofacial development. (**a**) Protocol used for induction of Cre-recombination and expression of tdTomato fluorescent protein (red) upon tamoxifen injection. (**b**) Fluorescent tdTomato on cryostat sections of calvaria (upper panel) at the postnatal P5, P14, and P60 showing labeling (white arrowheads) in suture mesenchyme, periosteum (Ps) and dura mater (Du), subsequently contributing to osteocytes (Ocy) (inset box) and bone marrow (Bm) of calvarial bones over time. Scale bar: 100 µm. Lower panel, cranial

*Figure 5 continued on next page*

*Figure 5 continued*

base synchondrosis (ISS) showing labeling at P5 in resting (RZ) and proliferative (PZ) chondrocyte zones, but not in the hypertrophic zone (HZ) (cyan dotted lines), consistent with X-gal staining. Lineage trace at P14 shows an increase in tdTomato + cells in all synchondrosis regions and associated bone marrow (Bm). At P60, tdTomato labeling is persistent in the middle zone and shows a clone of tdTomato labeling in proliferative and hypertrophic chondrocytes (yellow asterisk) and appears in the lining of bone marrow (Bm) and osteocytes (Ocy) of cortical bone (Ct). Scale bar: 100 µm (P14 and P60) and 50 µm (P5). Gray: cell nuclei. (**c**) Protocol used for induction of Cre-recombination and expression of tdTomato fluorescent protein (red) in *Gli1^CreERT^*; *Rosa26^LSL-tdTomato^* mice. (**d**) Fluorescent tdTomato (red) on cryostat sections of calvaria (right) and cranial base synchondroses (ISS and SOS, left) from 2-week-old *Gli1^CreERT^*; *Rosa26^LSL-tdTomato^* mice showing labeling in a similar cell population to that seen with *Ddr2^mer-iCre-mer^*. Scale bar: 200 µm (Suture; left) and 100 µm (ISS and SOS; right). Boxed region is shown in higher magnification (bottom). RZ: resting zone; PZ: proliferative zone; HZ: hypertrophic zone. (**e**) Representative immunofluorescence images showing *Gli1* (green) and *Ddr2* (red) immunostaining of coronal sutures, SOS and ISS from 2-week-old mice. Scale bar: 100 µm. Boxed region is shown in higher magnification (bottom). White arrowheads indicate co-localization. Cell nuclei were stained with DAPI (blue). Bm: Bone marrow; Tb; trabecular bone.

The online version of this article includes the following figure supplement(s) for figure 5:

**Figure supplement 1.** *Ddr2^mer-iCre-mer^* induced recombination in cranial sutures and cranial base synchondrosis.

**Figure supplement 2.** Co-localization of DDR2 and GLI1 in cranial suture and synchondroses in 2-week-old mice.

The cranial base synchondroses are a second major site of *Ddr2* expression. TdTomato labeling was examined in both ISS (*Figure 5b*) and SOS (*Figure 5—figure supplement 1b*). At day 5, TdTomato labeling was detected in some cells in resting and proliferative zones, where *Ddr2*-labeled cells were either single cells dominant in the resting zone or pairs of daughter cells dominant in proliferative zone. However, *Ddr2* showed no labeling in hypertrophic chondrocytes. These results are in agreement with *Ddr2-LacZ* expression, indicating that tdTomato-labeled cells include *Ddr2*-expressing cells. By two weeks, progeny of *Ddr2*-labeled cells were seen in all chondrocyte lineages and associated bone marrow. By 2 months, labeled cells persisted in the resting chondrocyte zone and formed columns of cells extending through proliferative and hypertrophic zones. Progeny of *Ddr2*-expressing cells were also seen on the surface of trabecular bone, in the bone marrow and in osteocytes within the cortical bone of the cranial base (*Figure 5b*, *Figure 5—figure supplement 1b*). Together, our results indicate that *Ddr2* is expressed in cell populations within cranial sutures and synchondroses having features of skeletal progenitor cells, but the identity of these cells still needs to be determined.

GLI1, a mediator of hedgehog signaling, is associated with stem/progenitor cells in cranial sutures (*Zhao et al., 2015*). We asked if there is overlap between *Ddr2*- and *Gli1*-expressing cells. To address this question, we verified the activity of inducible *Gli1-Cre^ERT^* under our experimental conditions by breeding with *Rosa26^LSL-tdCreERT^* reporter mice. Newborn mice heterozygous for inducible *Gli1^CreERT^* and the *Rosa26^LSL-tdTomato^* allele were given four intragastric tamoxifen injections and analyzed after 2 weeks (*Figure 5c*). As previously reported (*Zhao et al., 2015*), *Gli1^CreERT^* labeling was detected in cranial sutures in a pattern like that seen for *Ddr2* (*Figure 5d*). *Gli1^CreERT^* also showed labeling in all chondrocyte lineages of cranial base synchondroses, with high concentration in resting and proliferative chondrocytes (*Figure 5d*). This suggests that *Gli1* expression is not restricted to cells in cranial sutures or subchondral metaphyseal bone as previously reported, but it is also in growth plate chondrocytes of synchondroses. The similarity between *Ddr2^Mer-icre-Mer^* and *Gli1^CreERT^* labeling in cranial sutures and synchondroses suggests there may be a functional overlap between *Ddr2*- and *Gli1*- expressing cells. In support of this concept, we confirmed co-expression in the calvaria and synchondroses using co-immunostaining with DDR2 and GLI1 antibodies (*Figure 5e*, *Figure 5—figure supplement 2* for quantification of colocalization). A high degree of colocalization was observed in select suture cells and adjacent periosteum and in the central resting zone and hypertrophic regions of the SOS and ISS with a subfraction of GLI1 + cells (68–78 percent) also being DDR2+. The DDR2-GLI1 colocalization in synchondroses is reminiscent of what we previously observed in long bone growth plates where colocalization was preferentially seen in resting, proliferative and hypertrophic chondrocytes (*Mohamed et al., 2022*).

## *Ddr2* functions in Gli1+ skeletal progenitors to control craniofacial morphogenesis

Based on the aforementioned results, we conclude that global *Ddr2* deficiency alters craniofacial morphology by affecting both calvarial bone thickness and frontal suture mineralization as well as endochondral bone growth in the cranial base. Furthermore, our localization and lineage tracing

studies suggest preferential expression of *Ddr2* in suture-associated skeletal progenitors and resting/proliferating zone chondrocytes in cranial base synchondroses. Substantial overlap was noted between DDR2 and cells expressing the skeletal progenitor marker, GLI1. *Ddr2* expression was also detected in resting and proliferating chondrocytes where type 2 collagen is present. To determine if *Ddr2* has cell-autonomous functions in specific cell populations, a conditional deletion strategy was employed.

Our initial focus was on *Gli1*-expressing cells since this marker is associated with suture-associated skeletal progenitors as well as cranial base chondrocytes. *Gli1^CreERT* mice were crossed with mice carrying a floxed *Ddr2* allele, which includes exon 8 flanked by LoxP sites to generate conditional knockout mice (*Gli1^CreERT; Ddr2^fl/fl* mice). To disrupt the *Ddr2* gene, we injected *Gli1^CreERT; Ddr2^fl/fl* newborn mice and their control littermates (*Ddr2^fl/fl*) intragastrically with tamoxifen for four days and analyzed them at 3 months (*Figure 6a*). PCR analysis of genomic DNA extracted from ear punches confirmed the efficiency of recombination (*Figure 6b*). This was further validated by immunofluorescence of sections through the cranial base using an anti-pDDR2 antibody (*Figure 6k*). To investigate how *Ddr2* loss in *Gli1*-expressing cells affected the craniofacial skeleton, we performed AP linear skull measurements from 3-month-old mice as described in *Figure 1*. Tamoxifen-treated *Gli1^CreERT; Ddr2^fl/fl* mice had craniofacial abnormalities of similar type and magnitude to those seen in global *Ddr2* knockouts. Conditional knockout mice had significantly reduced skull length (11%) due to shortening in the nasal bones, cranial vault, and posterior cranial base (*Figure 6c–e*) and reduced anterior skull width in the absence of significant changes in skull height at most sites except posterior cranial height that was reduced by approximately 5% (*Figure 6f–g*). *Gli1^CreERT; Ddr2^fl/fl* mice also had thin frontal bones (*Figure 6d and h*). Also, like *Ddr2^slie/slie* mice, delayed mineralization in the posterior portion of the frontal suture was observed in 5 of 10 mutant skulls; this was not seen in *Ddr2^fl/fl* mice (n=10, *Figure 6D*). Analyzing 2-week-old skulls, *Gli1^CreERT; Ddr2^fl/fl* mice also had abnormally wide cranial base synchondroses with greatest changes seen in the ISS (*Figure 6i–k*). *Gli1-Cre^ERT* in the absence of the *Ddr2^fl/fl* allele did not affect synchondrosis morphology (*Figure 6i*). Similarly, mutant mice showed a significant shortening in cranial base bones, including presphenoid (13%), basisphenoid (~15%), and basio-occipital bones (11%), suggesting impaired endochondral ossification at 3 months (*Figure 6m–p*).

Together, these results are consistent with the GLI1 distribution we observed in cranial sutures and synchondroses and suggest that DDR2 functions in GLI1-positive cells to both control suture formation/calvarial bone mineralization as well as endochondral growth in the cranial base. However, it is also possible that changes in suture formation/cranial vault growth could be secondary to reduced growth of the cranial base as has been previously proposed (*Kreiborg et al., 1993*). If this were true, the cranial vault defects seen in *Ddr2^slie/slie* and *Gli1^CreERT; Ddr2^fl/fl* mice could be secondary to deficient growth at the cranial base rather than reflecting separate functions of *Ddr2* in sutures and calvarial bone. To discriminate between these possibilities, *Col2a1-Cre; Ddr2^f/f* mice were developed to selectively deleted *Ddr2* in chondrocytes (*Ovchinnikov et al., 2000*). At birth, mutant mice were viable and indistinguishable from their control littermates, but as they matured, *Col2a1-Cre; Ddr2^fl/fl* mice exhibited growth defects compared with littermate controls (*Ddr2^fl/fl*). The *Col2a1-Cre* transgene by itself did not cause any change in the cranial base synchondroses as demonstrated by whole mount staining (*Figure 7i*). Inhibition of *Ddr2* signaling in cranial base cartilage was confirmed by pDDR2 immunofluorescence (*Figure 7j*) as well as PCR analysis of DNA extracted from ear cartilage (not shown). Linear measurements on micro-CT scans of 3-month-old skulls (n=10) showed significantly reduced AP skull growth and increased anterior and posterior skull width and reduced posterior cranial height in *Col2a1-Cre; Ddr2^fl/fl* mice (*Figure 7a–d*). However, the reduction in AP skull length (7%) did not reach levels seen in *Ddr2^slie/slie* (12%) or *Gli1^CreERT; Ddr2^fl/fl* skulls (11%) (*Figure 1* **versus 6**). *Col2a1^Cre; Ddr2^fl/fl* mice also exhibited growth defects in endochondral bones, mainly BS and BO contributing to cranial base hypoplasia (*Figure 7h*). In addition, the ISS was significantly wider (270%) and higher (32%) (*Figure 7g and f*). The width and height of the SOS in *Gli1^CreERT;Ddr2^fl/fl* was also increased, but to a lesser extent than for the ISS (*Figure 7g*). This phenotype is like that seen in *Ddr2^slie/slie* and *Gli1^CreERT;Ddr2^fl/fl* mice. In contrast, conditional deletion of *Ddr2* in chondrocytes minimally affected the AP growth of cranial vault (decreased by 2.5%) (*Figure 7b*). *Col2a1^Cre; Ddr2^fl/fl* mice exhibited significant thinning in the frontal bone (56%) as compared with controls (*Figure 7a and e*). Unlike *Ddr2^slie/slie* or *Gli1^CreERT;Ddr2^fl/fl* mice, the parietal bone (11.4%,) and occipital bone (12%) also showed a moderate decrease in thickness-11.4 and 12%, respectively (*Figure 7e*). Significantly, conditional knockout in *Col2a1-Cre* lineages did not affect cranial sutures which were indistinguishable from *Ddr2^f/f* controls

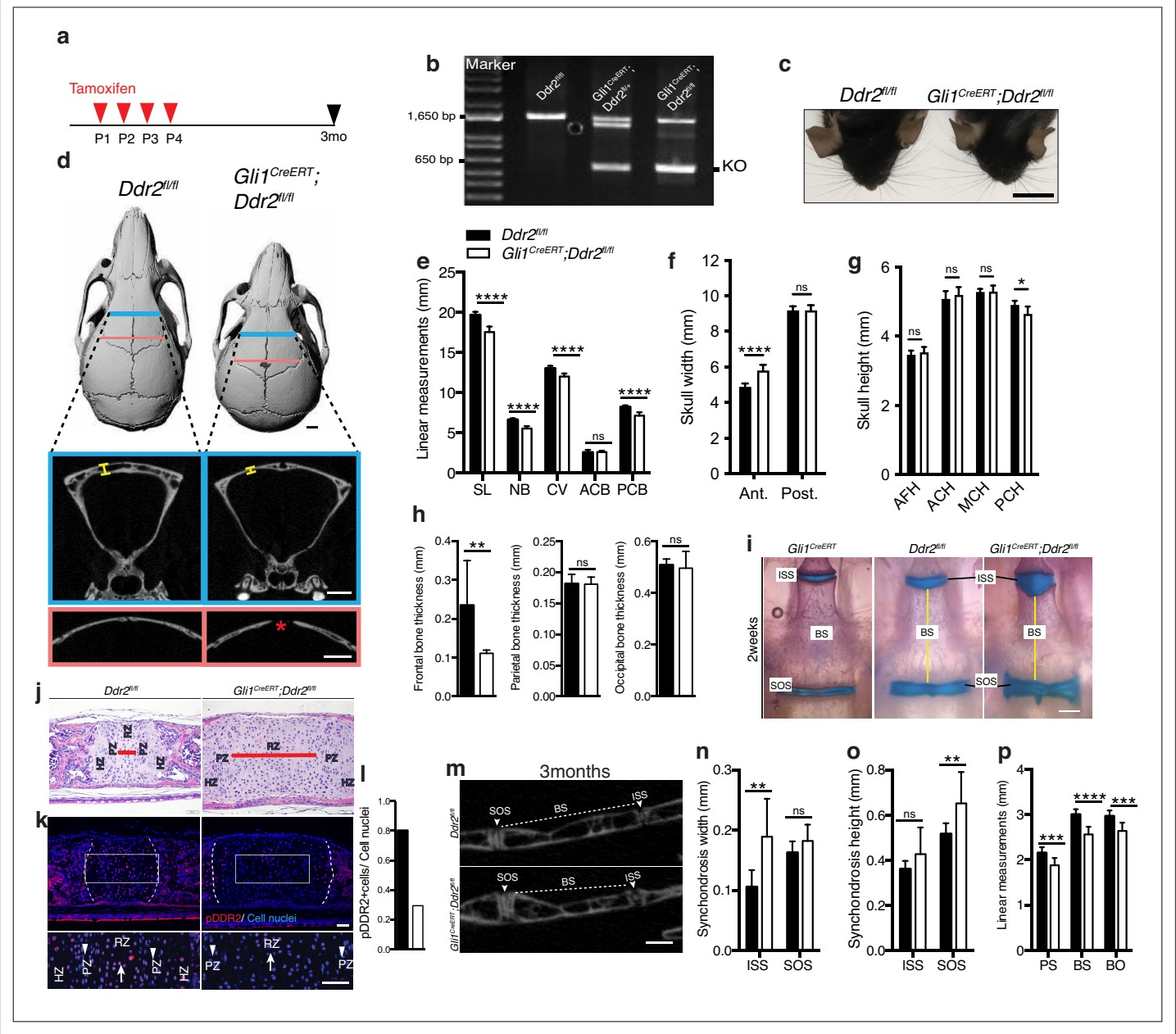

**Figure 6.** Loss of *Ddr2* in *Gli1*-expressing cells resulted in a craniofacial phenotype similar to *Ddr2^slie/slie^* mice. (**a**) Protocol used for induction of Cre-recombination upon tamoxifen injection. (**b**) Genotyping PCR showing WT (lower band) and *Ddr2* floxed alleles (upper band) near 1650 bp and recombined knockout allele below the 650 bp marker (KO). (**c**) Top head view showing *Gli1^CreERT^; Ddr2^fl/fl^* mice have short snout compared with *Ddr2^fl/fl^* mice. Scale bar: 1 cm. (**d**) μCT scans of *Ddr2^fl/fl^* and *Gli1^CreERT^; Ddr2^fl/fl^* skulls show reduced anterior-posterior skull length and increased anterior skull width (quantification in **e–g**). Note thinning and suture defect in the frontal bone in *Gli1^CreERT^; Ddr2^fl/fl^* skulls (**d**, bottom). Scale bar: 1 mm. (**h**) Quantification of frontal, parietal, and occipital bone thickness. (**i**) Alcian blue and Alizarin red whole mount staining shows *Gli1-Cre; Ddr2^fl/fl^* skulls have wide cranial base synchondroses compared with *Gli1^CreERT^ and Ddr2^fl/fl^*. Scale bar: 500 μm. (**j**) H&E staining of ISS shows widening of resting zone and chondrocyte disorganization in *Gli1^CreERT^; Ddr2^fl/fl^* mice. Scale bar: 50 μm. RZ: Resting zone; PZ: Proliferative zone; HZ: Hypertrophic zone. Red bar compares RZ width. (**k**) Immunofluorescence images show reduced pDDR2 (red) immunostaining indicative of reduced DDR2 signaling in *Gli1^CreERT^; Ddr2^fl/fl^* synchondrosis. Dotted lines denote chondro-osseous junction. Boxed region is shown in higher magnification, lower panel. Cell nuclei were stained with DAPI (blue). Arrows, resting zone; arrowheads, proliferative zone. Scale bar: 50 μm. (**l**), quantification of immunostaining in k. (**m-p**) μCT images and quantification show enlarged synchondroses associated with shortening in cranial base bone lengths in 3-month-old *Gli1^CreERT^; Ddr2^fl/fl^* skulls compared with controls. Scale bar: 500 μm. c-h, (**m-p**) 3-month-old mice. (**i–l**) 2-week-old mice. Data are presented as mean ± SD. (n=10). *p<0.01, **p<0.01, ***p<0.001, ****p<0.0001, ns, not significant, two-tailed unpaired *t* test.

The online version of this article includes the following source data for figure 6:

**Source data 1.** Skull linear measurements in *Ddr2^fl/fl^* versus *Gli1^CreERT^ ;Ddr2 ^fl/fl^* mice.

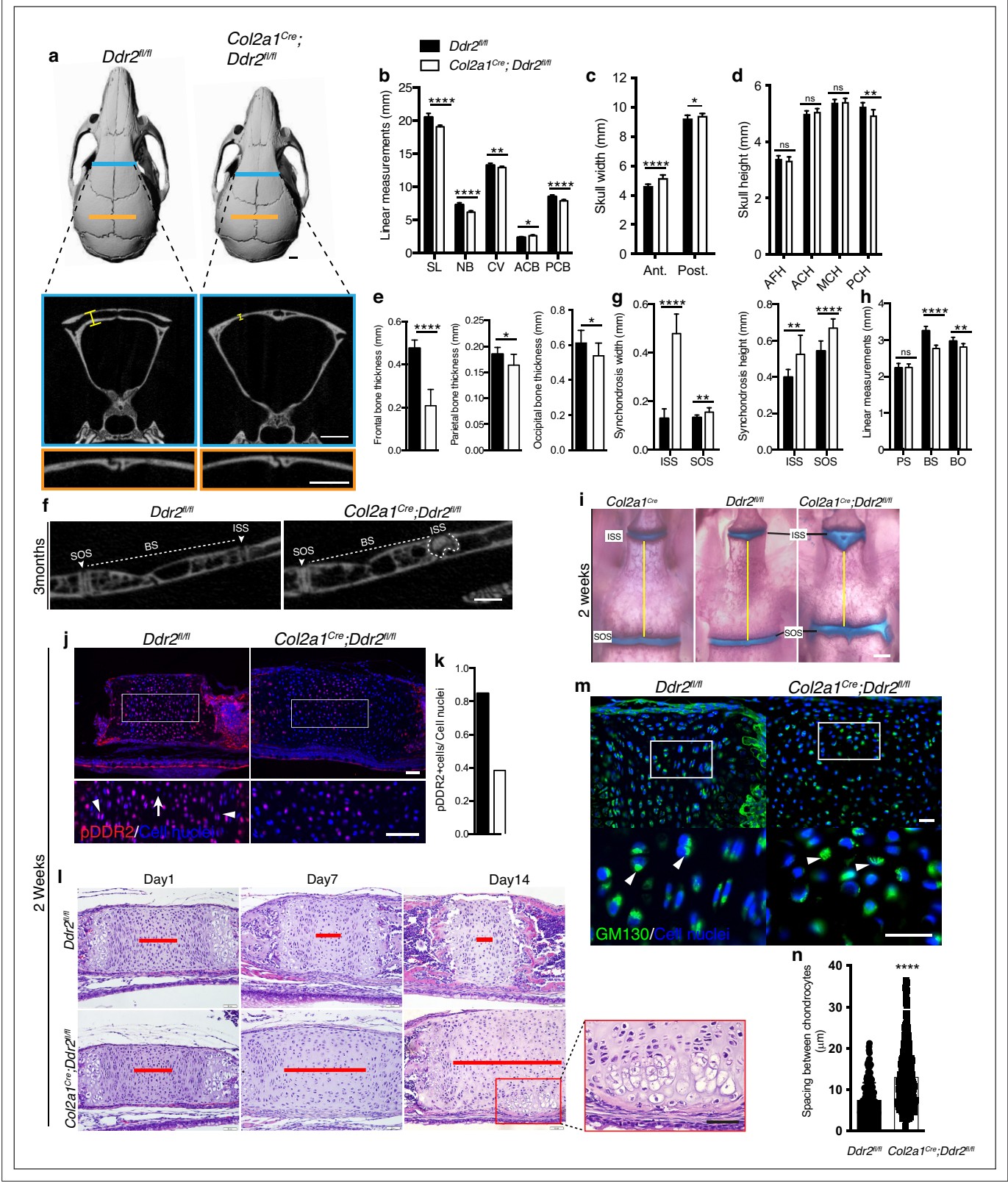

**Figure 7.** *Ddr2* conditional knockout in *Col2a1*-expressing chondrocytes causes cranial base hypoplasia and alters synchondroses without affecting cranial sutures. (**a-d**) µCT scans of *Ddr2^fl/fl^* and *Col2a1^Cre^;Ddr2^fl/fl^* skulls (3-month-old) showing reduced anterior-posterior skull length, length of individual bones and increased anterior and, to a lesser extent, posterior skull width in conditional knockout mice. Note thinning of frontal bone in *Col2a1^Cre^;Ddr2^fl/fl^* skulls, but no effect on cranial sutures. Scale bars: 1 mm. (**e**) Quantification of frontal, parietal, and occipital bone thickness. (**f–h**) Quantification of

*Figure 7 continued on next page*

*Figure 7 continued*

µCT scans showing enlarged synchondroses associated with shortening in cranial base bone lengths in *Col2a1^Cre^;Ddr2^fl/fl^* skulls. Data are presented as mean ± SD. (n=10). *p<0.01, **p<0.01, ***p<0.001, ****p<0.0001, ns, not significant, two-tailed unpaired *t* test. (**i**) Alcian blue and alizarin red whole mount staining showing 2-week *Col2a1^Cre^;Ddr2^fl/fl^* skulls had wide cranial base synchondroses compared with *Col2a1^Cre^* and *Ddr2^fl/fl^*. Scale bar: 500 µm. (**j**) Immunofluorescence images showing reduced pDDR2 (red) immunostaining indicating loss of DDR2 signaling in *Col2a1^Cre^;Ddr2^fl/fl^* mice (Quantification in **k**). Scale bar: 50 µm. Boxed region is shown at higher magnification, lower panel. Cell nuclei were stained with DAPI (blue). (**l-n**) Analysis showing *Col2a1^Cre^;Ddr2^fl/fl^* mice exhibited time-dependent widening in resting zone, altered polarization and ectopic hypertrophy. (**l**) Time-course analysis using H&E staining shows no difference in histological structures of the ISS between *Col2a1^Cre^;Ddr2^fl/fl^* mice and their control littermates at P1, but during the first 2 weeks, the resting zone became abnormally wide (red lines) and exhibited ectopic hypertrophy on the ventral side of cranial base synchondrosis (red box). Scale bar: 50 µm. (**m**) Immunofluorescent images of GM130 staining (green) shows well-defined Golgi staining adjacent to the nucleus of cells in RZ of wild type synchondroses, but in mutant synchondroses, GM130 immunostaining is diffuse and ill-defined indicating disturbed cell organization. Boxed region is shown in higher magnification, lower panel. Cell nuclei were stained with DAPI (blue). Scale bar: 20 µm. (**n**) Linear measurements shows increased spacing between chondrocytes in resting zone of wildtype and mutant synchondroses. Spacing between chondrocytes was measured by drawing lines between chondrocytes in resting zone using ImageJ. (**a–h**) 3-month-old mice, (**i–n**) 2-week-old mice. Data are presented as mean ± SD. (n=3). ****p<0.0001, two-tailed unpaired *t* test.

The online version of this article includes the following source data and figure supplement(s) for figure 7:

**Source data 1.** Skull linear measurements in *Ddr2^fl/fl^* versus *Col2a1^Cre^;Ddr2^fl/fl^* mice.

**Source data 2.** Measurement of spacing between chondrocytes in *Ddr2^fl/fl^* versus *Col2a1^Cre^;Ddr2^fl/fl^* mice.

**Figure supplement 1.** *Ddr2* loss in mature osteoblasts using *Bglap^Cre^* did not result in craniofacial abnormalities.

**Figure supplement 2.** Synchondroses in *Col2a1^Cre^:Ddr2ff* mutants exhibited deficient chondrocyte proliferation and abnormal type II collagen distribution.

(n=10 mice/genotype, *Figure 7a*). This clearly resolves functions of *Ddr2* in cranial base that were disrupted in *Ddr2^slie/slie^*, *Gli1^CreERT^;Ddr2^fl/fl^* and *Col2a1^Cre^; Ddr2^fl/fl^* mice from functions in sutures that were only disrupted in *Ddr2^slie/slie^* and *Gli1^CreERT^;Ddr2^fl/fl^* animals and establishes separate functions for *Ddr2* in both suture mesenchyme and cranial base chondrocytes.

In contrast to results with *Gli1^CreERT^;Ddr2^fl/fl^* and *Col2a1^Cre^;Ddr2^fl/fl^* mice, conditional deletion of *Ddr2* in mature osteoblasts using *Bglap^Cre^* did not affect skull length, cranial sutures, or the cranial base in 3-month-old mice (*Figure 7—figure supplement 1*). This Cre line induces efficient recombination in mature osteoblasts/osteocytes (*Zhang et al., 2002*) and we detected efficient excision of the floxed *Ddr2* allele in DNA from vertebrae-containing tail biopsies from *Bglap^Cre^;Ddr2^fl/fl^* mice (result not shown). These results are consistent with craniofacial functions of *Ddr2* being restricted to skeletal progenitors and chondrocytes rather than mature bone forming cells.

## Loss of *Ddr2* in chondrocytes causes ectopic hypertrophy and disrupted cell polarization

The presence of abnormal synchondroses and associated cranial base hypoplasia was a consistent finding after global or conditional *Ddr2* inactivation. In all cases, reduced growth of BS and BO bones was observed and chondrocyte organization into resting, proliferative and hypertrophic zones was disrupted. To further define the onset and evolution of synchondrosis changes in *Ddr2* deficiency, we conducted a time-course histological analysis focusing on ISS morphology of *Col2a1^Cre^;Ddr2^fl/fl^* mice (*Figure 7l*). At birth, no significant differences in synchondroses were seen between mutant and control littermates. However, as endochondral ossification progressed, the resting zone (highlighted with red lines) became progressively narrower in controls while expanding at the expense of the proliferative and hypertrophic zones in *Col2a1^Cre^;Ddr2^fl/fl^* mice (*Figure 7l*). As was the case in *Ddr2^slie/slie^* mice (*Figure 2*), synchondrosis widening could not be explained by increased cell proliferation which was reduced in ISS and SOS (*Figure 7—figure supplement 2a*). The question became why mutant synchondroses were enlarged and exhibited an increase in the cell number, particularly in the central resting zone (*Figure 7l*). It is possible that chondrocytes proliferating at a lower rate were retained in the resting zone rather than transitioning to the proliferative zone. Also, as was seen in *Ddr2^slie/slie^* mice, type II collagen distribution was abnormal in *Col2a1^Cre^;Ddr2^fl/fl^* mice with a selective loss of staining in the inter-territorial matrix. In *Col2a1^Cre^;Ddr2^fl/fl^* mice, we also observed an increase in average spacing between cells (*Figure 7—figure supplement 2c*, *Figure 7n*). Together, these changes likely contributed to the observed widening in the synchondrosis resting zone.

Intriguingly, P14 *Col2a1^Cre;Ddr2^fl/fl* mice formed an ectopic hypertrophic zone on the ventral side of the ISS at right angles to the normal synchrondrosis axis. This hypertrophic region was in addition to the normal hypertrophic zones on either end of synchondroses (*Figure 7l*). MicroCT images of 3-month-old mice also showed W-shaped synchondroses in *Col2a1^Cre;Ddr2^fl/fl* mice (*Figure 7f*), suggesting ectopic endochondral ossification taking place in the ventral side of the ISS. This phenotype was seen in 8 out of 10 mutant mice examined. With the shift in chondrocyte organization, we asked if cell polarity was disrupted in the absence of *Ddr2*. To test this, we examined the cellular distribution of GM130, a Golgi apparatus marker that normally maintains a fixed orientation to the nucleus in chondrocytes (*Arita et al., 2015*). In control mice, immunostaining of GM130 showed well-defined, localized staining adjacent to the nucleus, indicating that positional information was retained within the growth plate (*Figure 7m*). In mutant synchondroses; however, GM130 immunostaining was disorganized, suggesting that chondrocyte polarization was disrupted. This was also seen in *Ddr2^slie/slie* and *Gli1^CreERT; Ddr2^fl/fl* synchondroses (data not shown). Together, our findings indicate that proper DDR2 signaling is important for specification of cell polarity possibly through interaction with the cartilage extracellular matrix; however, a direct role of *Ddr2* on chondrocyte polarity cannot be excluded.

## Discussion

The ECM is a critical regulator of skeletal development and has been implicated in a wide spectrum of human skeletal disorders, many of which involve craniofacial structures (*Lu et al., 2011*; *Erlebacher et al., 1995*; *Velleman, 2000*; *Kuivaniemi et al., 1997*). In this study, we highlight the importance of collagen-cell interactions with emphasis on cell-autonomous functions of the cell surface collagen receptor, DDR2, during growth of the craniofacial skeleton. Characterization of craniofacial phenotypes of globally *Ddr2*-deficient mice revealed two types of defects; (1) abnormalities in development of the flat bones of the skull including defective frontal suture formation/mineralization and thinning of frontal bones, (2) reduced anterior-posterior skull growth related to delayed endochondral ossification at cranial base synchondroses. To understand the basis for these defects, we conducted a systematic characterization of the distribution and lineage of *Ddr2*-expressing cells and used a conditional deletion approach to resolve the cellular sites of action of *Ddr2*. These studies complement our previous analysis of DDR2 function in development of the appendicular skeleton (*Mohamed et al., 2022*).

As a prerequisite for understanding function, we determined the spatial distribution of *Ddr2* expression during craniofacial development. Analysis of LacZ knock-in reporter mice revealed expression in developing midface as early as E11.5 with staining initially seen in cartilage primordia of the cranial base, nasal septum, and Meckel's cartilage in developing mandible. Later during development, *Ddr2* expression was localized in neonatal calvariae in developing periosteum, dura mater, and cranial sutures where craniofacial stem cells reside. In the adult, it was also localized to cranial sutures, periosteum, dura mater and in bone marrow lining cells. *Ddr2* expression was either low or undetected in osteocytes. *Ddr2* expression was also localized to the resting and proliferative zones of neonatal cranial base synchondroses but was not detected in hypertrophic chondrocytes. These findings led us to hypothesize that *Ddr2* regulates development of the craniofacial skeleton through its function in skeletal progenitor cells and chondrocytes.

This concept is further supported by lineage tracing studies conducted in *Ddr2^Mer-icre-Mer;Rosa26^LSL-tdTomato* mice where an initial tamoxifen pulse labeled cells in cranial suture mesenchyme, periosteum and dura as well as resting and proliferating cells of synchondroses. Over a two-month chase period, the TdTomato label increased in sutures and expanded into lining cells of bone marrow and calvarial osteocytes while, in the cranial base, labeled cells persisted in the central resting zone and expanded to proliferative and hypertrophic zones of synchondroses and into adjacent bone. While these studies do not prove *Ddr2 i*s expressed in stem/progenitor populations, they show DDR2-positive cells can serve as precursors to cells of the bone lineage. Significantly, in related studies reported in preprint form, a DDR2 +cranial suture cell population was described (*Greenblatt et al., 2021*) that is distinct from previously characterized CTSK +suture stem cells (*Debnath et al., 2018*). These DDR2 + cells have several stem cell properties including capacity for self-renewal, tri-lineage differentiation potential to osteoblasts, adipocytes and chondrocytes, expression of several SC markers including GLI1 and capacity to generate all DDR2 + cells present in the native suture. However, further studies will be required to determine whether the DDR2 + cells we identified in synchondroses also share these stem cell properties.

Consistent with DDR2 functioning in skeletal progenitors, we observed substantial colocalization of DDR2 with a subset of GLI1 + cells in calvarial sutures, periosteum and synchondroses. GLI1-positive sutural stem cells (SuSCs) are of great interest because of their contribution to the development and regeneration of craniofacial bones and possible role in craniofacial abnormalities. Ablation of GLI1 +SuSCs leads to craniosynostosis, a premature closure of cranial sutures resulting in arrested skull growth (*Zhao et al., 2015*; *Durham et al., 2019*), while impaired cell proliferation and osteogenic differentiation of SuSCs is associated with wide-open fontanels due to reduced or delayed bone growth (*Goto et al., 2004*; *Qin et al., 2019*). To determine if DDR2 functions in GLI1 + cells, we conditionally inactivated *Ddr2* using *Gli1^{CreERT}; Ddr2^{fl/fl}* mice. Loss of *Ddr2* in *Gli1*-expressing cells faithfully recapitulated the cranial phenotype of *Ddr2^{slie/slie}* mice including the presence of open frontal sutures (analogous to the metopic suture in humans) and thinning of frontal bones. This phenotype also mimics the open fontanels seen in SMED, SL-AC patients. These results are consistent with a function for *Ddr2* in Gli1 +sutural cells to regulate osteoblast differentiation during intramembranous bone formation.

Intriguingly, our lineage tracing studies identified *Gli1^{CreERT}*-induced tdTomato labeling beyond previously reported sites in cranial sutures, long bone growth plates and metaphyses (*Zhao et al., 2015*; *Mohamed et al., 2022*; *Shi et al., 2017*). Specifically, this *Cre* also marked chondrocyte lineages and co-localized with *Ddr2* in cranial base synchondroses. Consistent with this distribution, tamoxifen treated *Gli1^{CreERT}; Ddr2^{fl/fl}* mice had cranial base defects that were similar in type and magnitude to those seen in *Ddr2^{slie/slie}* mice. Since it has been proposed that changes in the cranial base can indirectly affect suture formation (*Kreiborg et al., 1993*), we conducted a second conditional knockout of *Ddr2* in chondrocytes using *Col2a1^{Cre}* that would be expected to be preferentially active during endochondral bone formation in synchondroses, but not during intramembranous bone formation in sutures (*Figure 3*). The synchondroses of *Col2a1^{Cre};Ddr2^{f/f}* and *Gli1^{CreERT};Ddr2^{fl/fl}* mice were similar. However, *Col2a1^{Cre};Ddr2^{f/f}* mice had normal frontal sutures. Therefore, the contribution of *Ddr2* to endochondral bone formation in the synchondrosis is independent from its function in cranial sutures.

An unexpected finding was the observed thinning of frontal bones in *Col2a1^{Cre};Ddr2^{f/f}* mice. This thinning, which was first detected in *Ddr2^{slie/slie}* and *Gli1^{CreERT};Ddr2^{fl/fl}* mice, would not normally be expected at a site of intramembranous bone formation that is not a major site of type 2 collagen synthesis. However, interpretation of this result is confounded by earlier studies showing endogenous *Col2a1* expression and *Col2a1^{Cre}* activity can be detected in calvarial osteoblasts and suture mesenchymal cells (*Szabova et al., 2009*; *Sakagami et al., 2017*). It is therefore possible that this *Cre* could inactivate *Ddr2* in a select population of *Col2a1*-expressing cells in intramembranous bone, thereby explaining the reduced frontal bone thickness in *Col2a1^{Cre};Ddr2^{fl/fl}* mice. However, the absence of a posterior frontal suture defect in *Col2a1^{Cre};Ddr2^{fl/fl}* mice argues against *Ddr2* having a function in *Col2a1*-expressing cells necessary for normal suture formation.

Significant similarities and differences are apparent when results of the present study are compared with our previous work on DDR2 functions in the appendicular skeleton (*Mohamed et al., 2022*). At both sites, DDR2 was localized to resting and proliferative zone cartilage of synchondroses or long bone growth plates, respectively, where co-localization with the Hh intermediate, GLI1, was observed. Lineage tracing at both sites suggested that DDR2 positive cells could serve as progenitors for hypertrophic chondrocytes, osteoblasts, and osteocytes of endochondral bone. Consistent with this cartilage localization, inactivation of *Ddr2* in Gli1-positive progenitors (*Gli1^{CreERT}*) or in chondrocytes (*Col2a1^{Cre}*) inhibited endochondral bone formation in synchondroses and tibial growth plates and this inhibition was related to inhibition of chondrocyte proliferation rather than an increase in apoptosis. Type II collagen matrix organization and chondrocyte alignment was also disrupted at both sites although this phenotype was more severe in the synchondrosis where chondrocytes formed an ectopic hypertrophic zone. DDR2 functions in endochondral bone formation likely explain the dwarf phenotype, reduced A-P skull length and shortened snout of *Ddr2* deficient mice and as well as the dwarfism and the depressed facial features of SMED, SL-AC patients. In mice, *Ddr2* was also expressed adjacent to intramembranous bone in the periosteum of calvaria and tibial cortical bone as well as cranial sutures. However, important differences in intramembranous bone formation were seen when comparing these two sites. In the cranial vault, *Ddr2* deficiency inhibited frontal suture formation and reduced frontal bone thickness. In contrast, *Ddr2* deficiency had little effect on thickness of tibial cortical bone, which also forms through an intramembranous process (*Mohamed et al., 2022*).

The basis for these differences is not clear. On one hand, differences in rates of formation of calvarial versus cortical bone may mask DDR2-related activity in cortical bone. Alternatively, they may reflect inherent differences between these two tissue sites. Functions of DDR2 in cortical bone may be more apparent in regeneration models which involve more rapid bone formation than is seen during development. As we recently showed, *Ddr2* deficiency severely disrupts intramembranous bone healing of subcritical-sized calvarial defects (*Binrayes et al., 2021*). It is not known if healing of related cortical bone defects would also be disrupted.

Although the basis for DDR2 actions on cranial and skeletal development are not well-understood, we hypothesize that it may function by controlling properties of the ECM such as collagen distribution, fibril orientation and stiffness, which in turn alters cell behavior via regulation of integrin signaling. In the present study, we evaluated type II collagen properties in synchondroses using immunofluorescence and SHG microscopy. In wild type mice, uniform Col II staining in the extraterritorial and territorial chondrocyte matrix and a high degree of collagen orientation (increased anisotropy) was observed throughout the synchondrosis resting zone. However, Ddr2-deficient mice showed a non-homogenous Col II staining pattern, predominantly in pericellular space adjacent to chondrocytes as well as decreased anisotropy. These ECM changes were accompanied by decreased chondrocyte polarization as measured by GM130 Golgi staining, reduced proliferation, and formation of an ectopic hypertrophic zone. GM130 is a peripheral membrane protein tightly bound to the cytoplasmic face of the Golgi apparatus, the central organelle for secretory protein trafficking and cargo modification (*Arita et al., 2015*). GM130 is required for maintaining Golgi structure, spindle assembly, cell cycle progression, and cytoskeletal dynamics (*Wei et al., 2015*; *Sanders and Kaverina, 2015*; *Colanzi and Corda, 2007*; *Farquhar and Palade, 1998*; *Liu et al., 2017*). Moreover, disruption of GM130 in mitosis impairs spindle assembly and cell division (*Wei et al., 2015*). This may explain the underlying deficient chondrocyte proliferation seen in *Ddr2*-deficient synchondroses, although these mechanisms require more clarification. In our previous study on long bone development, comparable changes in Col II distribution and chondrocyte proliferation were observed in the tibial growth plate (*Mohamed et al., 2022*).

In other experimental systems, similar DDR2-dependent changes in collagen fibril orientation and distribution have been observed and shown to modify cellular behavior by altering integrin signaling. For example, DDR2 in cancer-associated fibroblasts (CAFs) increases tumor stiffness by reorganizing type I collagen fibrils at the tumor-stromal boundary, thereby promoting lung metastasis of breast tumors. These ECM changes are associated with increased integrin activity and mechanotransduction by CAFs (*Bayer et al., 2019*). Also, in a recent collaborative study on trauma-induced heterotopic ossification (HO) in mice, we showed that loss of DDR2 was associated with decreased collagen fibril orientation as measured by SHG as well as dramatic inhibition of HO (*Pagani et al., 2022*). The loss of collagen alignment was accompanied by a reduction in integrin signaling as measured by reduced FAK phosphorylation, decreased nuclear translocation of TAZ and reduced YAP/TAZ signaling, consistent with DDR2 being required for force transduction, subsequent osteoblast differentiation and HO.

Further evidence that changes in Col II orientation/distribution might explain the cartilage phenotype of *Ddr2*-deficient mice comes from studies with *Col2a1* mutant mice (*Brown et al., 1981*; *Pace et al., 1997*; *Seegmiller et al., 1971*; *Spranger et al., 1994*; *Vikkula et al., 1994*). Both *Col2a1expDmm/Dmm* mice harboring a mutation in the C-propeptide globular domain of type II collagen (*Col2a1*) (*Pace et al., 1997*) as well as mice containing an R992C mutation associated with spondyloepiphyseal dysplasia *Arita et al., 2015* have defects in the assembly and folding of type II collagen, that alter collagen distribution/orientation (*Arita et al., 2015*; *Brown et al., 1981*; *Pace et al., 1997*). In both cases, mutant mice have a similar phenotype to Ddr2-nulls; they are dwarf, have a shortened vertebral column and small rib cage as well as craniofacial defects characterized by skull dysmorphogenesis, reduced skull length, reduced cranial base length, cleft palate, and short mandible. Interestingly, mice harboring the R992C mutation also exhibit disrupted chondrocyte polarity (*Arita et al., 2015*). Most importantly, the altered distribution of cartilage matrix was also reported in a SMED, SL-AC patient (*Borochowitz et al., 1993*). Histological analysis of the resting zone of costochondral cartilage from a SMED, SL-AC patient showed abnormally distributed chondrocytes with noticeable dark staining in surrounding matrix filling lacunar space (*Borochowitz et al., 1993*). The altered distribution of cartilage matrix highlights the similarity between histopathological features of the patient samples and our results in *Ddr2*-deficient mice, and provides important insights into

potential mechanisms underlying *Ddr2* regulation of bone development mediated by changes in ECM organization.

In summary, this study establishes a critical function of *Ddr2* in skeletal progenitor cells, including GLI1 + cells, and chondrocytes to control bone formation and cartilage growth during postnatal growth of the craniofacial skeleton, and advances our understanding of the roles of cell-matrix interactions in craniofacial morphogenesis. Our findings provide further insight into the cellular basis of cranial defects in SMED SL-AC patients as well as evolutionary changes in craniofacial structure related to altered DDR2 activity.

# Materials and methods

## Key resources table

| Reagent type (species) or resource | Designation | Source or reference | Identifiers | Additional information |
|---|---|---|---|---|
| Gene (*Mus musculus*) | Discoidin Domain Receptor2 (*Ddr2*) | Genbank | Gene ID: 18214 | |
| Strain, strain background (*Mus musculus*, C57BL/6 J) | *Ddr2^slie/slie^* | Jackson lab | JAX:008172 | Breeding with C57BL/6 J mice at least 8 generations. |
| Strain, strain background (*Mus musculus*, C57BL/6 J) | *Ddr2^LacZ^* mice | Generated from 'knockout-first' *Ddr2^tm1a(EUCOMM)Wtsi^* -ES cells(European Mutant Mouse Repository) | EPD0607__B01 PMID:35140200 | 'knockout-first' mice crossed with Sox2Cre mice |
| Strain, strain background (*Mus musculus*, C57BL/6 J) | *Ddr2^fl/f^* | Generated from 'knockout-first' *Ddr2^tm1a(EUCOMM)Wtsi^* -ES cells(European Mutant Mouse Repository) | EPD0607__B01 PMID:35140200 | 'knockout-first' mice crossed with FlpO mice |
| Strain, strain background (*Mus musculus*, C57BL/6 J) | *Ddr2^mer-iCre-mer^* | Generated in UCSD Transgenic Animal Model Core and Embryonic Stem Cell shared resource | PMID:35140200 | |
| Strain, strain background (*Mus musculus*, C57BL/6 J) | *Rosa26^LSL-tdTomato^* | Jackson lab | JAX:007914 | |
| Strain, strain background (*Mus musculus*, C57BL/6 J) | *Gli1^CreERT^* | Jackson lab | JAX:007913 | |
| Strain, strain background (*Mus musculus*, C57BL/6 J) | *Col2a1^Cre^* | Generated at Richard R. Behringer lab | PMID:10686612 | |
| Strain, strain background (*Mus musculus*, C57BL/6 J) | *Bglap^Cre^* | Generated at Thomas L. Clemens lab | PMID:12215457 | |
| Sequence-based reagent | *Ddr2^fl/f^* _F | Renny T. Franceschi lab | PMID:35140200 | |
| Sequence-based reagent | *Ddr2^fl/f^* _R | Renny T. Franceschi lab | PMID:35140200 | |
| Sequence-based reagent | *Ddr2-LacZ*-F | Renny T. Franceschi lab | PMID:35140200 | |
| Sequence-based reagent | *Ddr2-LacZ*-R | Renny T. Franceschi lab | PMID:35140200 | |
| Sequence-based reagent | *Ddr2^mer-iCre-mer^* -F | Renny T. Franceschi lab | PMID:35140200 | |
| Sequence-based reagent | *Ddr2^mer-iCre-mer^* -R | Renny T. Franceschi lab | PMID:35140200 | |
| Antibody | Anti-human/mouse DDR2(Rabbit polyclonal) | LS Bio | LS-B15752 | IF(1:200) |

*Continued on next page*

*Continued*

| Reagent type (species) or resource | Designation | Source or reference | Identifiers | Additional information |
|---|---|---|---|---|
| Antibody | Anti-human/mouse Y740P-DDR2(Rabbit monoclonal) | R&D systems | MAB25382 | IF(1:200) |
| Antibody | Anti-Human/mousecleaved caspase 3(Rabbit polyclonal) | Cell Signaling | 9661 | IF(1:200) |
| Antibody | Anti-Human/mouse COL2(Rabbit polyclonal) | ABCam | Ab34712 | IF(1:100) |
| Antibody | Anti-Human/mouse GLI1(Rabbit polyclonal) | Novus Biologicals | NBP1-78259 | IF(1:100) |
| Antibody | Anti-Mouse IBSP(Rabbit polyclonal) | Renny T Franceschi lab | | IF(1:200) |
| Antibody | Anti-Human/mouse COL10(Rabbit polyclonal) | ABCam | Ab58632 | IF(1:100) |
| Antibody | Anti-Mouse COL1(Rabbit polyclonal) | Millipore | AB765P | IF(1:100) |
| Antibody | Anti-Mouse GM130(Mouse polyclonal) | BD Biosciences | 610822 | IF(1:100) |
| Commercial assay or kit | Click-iT EdU Alexa Fluor 488 Imaging Kit | Invitrogen | C10337 | |
| Commercial assay or kit | FragEL DNA Fragmentation Detection Kit | CalBiochem | QIA33-1 | |
| Chemical compound, drug | Tamoxifen | Renny T Franceschi lab | PMID:35140200 | |
| Chemical compound, drug | Aqua-Poly/Mount | Polysciences Inc | 18606 | |
| Chemical compound, drug | ProLong Gold Antifade Mountant with DAPI | Life technologies | P36931 | |
| Software, algorithm | Prism 9 | GraphPad | GraphPad Software, San Diego, CA | https://www.graphpad.com/ |
| Software, algorithm | ImageJ | ImageJ FibrilTool-Plugin | PMID:24481272 | https://imagej.nih.gov/ij/ |
| Other | Scano µCT 100 | MicroView software version 2.5.0 | PMID:35140200 | Methods-Skull morphometric analysis |

## Mice

*Smallie* mice (*Ddr2*<sup>slie/slie</sup>) (**Kano et al., 2008**) were obtained from Jackson laboratory (stock no. 008172). *Ddr2*<sup>fl/fl</sup> mice with *loxP* sites flanking coding exon 8 of *Ddr2* gene, *Ddr2-LacZ* mice and *Ddr2*<sup>mer-iCre-mer</sup> mice harboring a MerCreMer cassette knocked in-frame into exon 2 of the *Ddr2* locus were previously described (**Mohamed et al., 2020**; **Mohamed et al., 2022**). For lineage analysis, *Rosa26*<sup>LSL-tdTomato</sup> mice (**Madisen et al., 2010**) were crossed with *Ddr2*<sup>mer-iCre-mer</sup> heterozygous mice to generate *Ddr2*<sup>mer-iCre-mer</sup>; *Rosa26*<sup>LSL-tdTomato</sup> mice. Tamoxifen was administered as previously described (**Mohamed et al., 2022**).

To investigate the role of *Ddr2* in craniofacial development, we generated conditional knockout mice by crossing *Ddr2*<sup>fl/fl</sup> mice with tissue specific Cre driver mouse lines: *Col2a1*<sup>Cre</sup> (gift from Dr. Ernestina Schipani, U Michigan) (**Ovchinnikov et al., 2000**); *Gli1*<sup>CreERT</sup> (gift from Dr. Yuji Mishina, U. Michigan) (**Ahn and Joyner, 2004**); *Bglap*<sup>Cre</sup> (gift from Dr. Kurt Hankenson, U. Michigan) (**Zhang et al., 2002**). Mice used in our experiments were analyzed on C57BL/J6 background, except *Col2a1*<sup>Cre</sup>;*Ddr2*<sup>fl/fl</sup> mice were analyzed on mixed background.

All mice were housed under a 12 hr light cycle in compliance with the Guidelines for the Care and Use of Animals for Scientific Research. All protocols for mouse experiments were approved by the Institutional Animal Care and Use Committee of the University of Michigan. Genotyping of *Ddr2*<sup>mer-iCre-me</sup>, *Col2a1*<sup>Cre</sup>, *Gli1*<sup>CreERT</sup>, *Bglap*<sup>Cre</sup>, and *Rosa26*<sup>LSL-tdTomato</sup> mice mice was performed using PCR primers,

as previously described (*Ovchinnikov et al., 2000*; *Zhang et al., 2002*; *Madisen et al., 2010*; *Ahn and Joyner, 2004*). The genotyping of *Ddr2slie/slie* mice was performed using qRT-PCR with TaqMan probes on an ABI 7500 thermocycler (Applied BioSystems; *Ge et al., 2016*).

## Morphometric analysis of skulls

The craniofacial skeleton was examined by whole-mount skeletal staining as described previously (*McLeod, 1980*). For microcomputed tomography analysis, 10 skulls from 3-month-old male and female mice were harvested and fixed in 10% formalin overnight at 4 °C. Using microCT Scanco Model 100 (Scanco Medical), mouse skulls were scanned and reconstructed with voxel size of 12 µm, 70 kVp, 114 µA, 0.5 mm aluminum filter, and integration time of 500ms. For craniofacial characterization, skull scans in VFF files were reoriented and cropped in MicroView software version 2.5.0 (see orientation in *Figure 1—figure supplement 1*) and linear craniofacial measurements were made on 2D reoriented CT images using ImageJ software (version 1.51). We conducted skull linear measurements using previously published craniofacial landmarks (*Vora et al., 2015*). Thickness of cranial bones was measured on 2D reoriented CT images using a defined position on orthogonal views of MicroView software. Mid-bone regions (highlighted in blue and orange colors in *Figure 1b*) were selected to measure the thickness the frontal and parietal bones on the coronal (frontal) plane, where right and left measurements were taken around frontal and sagittal sutures for frontal and parietal bone, respectively, and the average of two measurements were reported. Thickness measurements of the occipital bone were made on the sagittal plane as shown in *Figure 1b and f*. Measurements for cranial base synchondroses were made on the mid sagittal plane, where synchondrosis width was determined by drawing a straight line in between chondro-osseous junctions with flanking bones and synchondrosis height determined by measuring the distance between two parallel lines tangential to the ventral and dorsal surfaces of synchondroses (*Figure 2a*). The length of cranial base bones was quantified by measuring the distance between two lines defining anterior and posterior borders of individual cranial base bones. All linear measurements were made on reoriented 2D µCT images using straight-line function in ImageJ.

## Histology and immunostaining

The whole skulls were fixed in 4% paraformaldehyde (PFA) for 48 hr at 4 °C. Specimens were decalcified in 10% ethylenediaminetetracetic acid (EDTA) (pH 7.2) and were then processed for paraffin embedding. Specimens were sectioned at 5 µm, deparaffinized and hydrated in ethanol series (100%, 95%, 70%) and in distilled water. For histological analysis, the sections were stained with hematoxylin and eosin according to the standard procedures. For immunofluorescence, sections were subjected to heat-induced antigen retrieval using 1 X Diva Decloaker (Biocare medical) following the manufacturer' instructions, washed with 1 X PBS and then incubated with blocking buffer containing 5–10% normal donkey serum, 1% bovine serum albumin (BSA), 0.01% tween in 1 X BSP for 1 hr at room temperature in a humid box. After blocking, the sections were incubated at 4 °C overnight, with the following primary antibodies: anti-DDR2 (LS B15752, 1:200), anti-Y740-P-DDR2 (R&D Systems MAB25382, 1:200), anti-cleaved caspase 3 (Cell Signaling 9661, 1:200), anti-COL2 (Abcam ab34712, 1:100); anti-COL10 (Abcam ab58632, 1:100); Anti-COL1(Millipore Sigma AB765P, 1:100); Anti-GM130 (BD Biosciences 610822,1:100); anti-IBSP (1:100). Anti-IBSP antibody was generated from a GST fusion protein containing amino acids 8–324 generated in the project laboratory. This was used as antigen for antibody production in rabbits (Harlan Laboratories). For Anti-GLI1 (Novus biological NBP1-78259, 1:100), sections were retrieved using citrate buffer (target retrieval solution, Dako) heated to 60 °C for 1 hr. The slides were rinsed three times with 1 X PBS, and the coverslip was mounted using ProLong Gold Antifade Mountant with DAPI (Life technologies) for cell nuclei staining. The sections were then imaged with a Nikon Eclipse 50i microscope and an Olympus DP72 camera.

## X-Gal (β-galactosidase) staining

X-Gal staining of heterozygous Ddr2-LacZ (Ddr2$^{+/LacZ}$) skulls was performed according to standard protocols (*Nagy et al., 2007*). Samples were fixed in 2% paraformaldehyde and 0.2% glutaraldehyde in 0.1 M phosphate buffer pH 7.3 plus 5 mM EGTA and 2 mM MgCl$_2$ at 4 °C. For the whole-mount staining, samples were rinsed after fixation three times in 1 X PBS plus MgCl$_2$, and then incubated at 37 °C overnight in a freshly prepared X-gal solution containing an X-gal substrate (UltraPure X-Gal,

Invitrogen), 2 mM MgCl$_2$, 5 mM potassium ferricyanide (III) (702587, Sigma), 5 mM potassium hexacyanoferrate (II) (P3289, Sigma), 0.01% sodium deoxycholate, 0.02% NP-40. The samples were then visualized using a dissection microscope (Nikon SMZ 745T), and images were captured using a Nikon DS-fi1 camera. The wild-type littermates were used as controls. For frozen sections, samples were decalcified with 20% EDTA (pH 7.2) for up to 2 weeks according to the mouse age, rinsed in 1 X PBS, then placed in 30% sucrose in 0.1 M phosphate buffer with 2 mM MgCl$_2$ at 4 °C for overnight. Samples were then embedded in optimal cutting temperature compound (Tissue-Tek) and cryosectioned at 12 μm thickness at –20 °C, mounted on glass histological slides (Fisherbrand ColorFrost Plus). For LacZ staining of frozen sections, sections were post-fixed in 0.2% PFA in 0.1 phosphate buffer pH 7.2 for 10 min on ice, washed 3 times in MgCl$_2$-containing PBS, and stained with an X-gal substrate (UltraPure X-Gal, Invitrogen), 2 mM MgCl$_2$, 5 mM potassium ferricyanide (III) (702587, Sigma) and 5 mM potassium hexacyanoferrate (II) (P3289, Sigma) overnight at 37 °C. The sections were washed 3 times in 1 X PBS followed by distilled water, counterstained with Vector Nuclear Fast Red staining, and dehydrated through ethanol series and xylene and mounted with an Acrytol mounting medium (Leica).

## Proliferation and apoptosis assay

To assay cell proliferation, mice were injected intraperitoneally with 5-ethynyl-2'-deoxyuridine and sacrificed 4 hr after injection. EdU-labeled cells were detected using Click-iT EdU Alexa Fluor 488 Imaging Kit (Invitrogen, # C10337). Briefly, after deparaffinization and hydration, the tissue sections were incubated with Click-it reaction mixture for 30 min in a dark humidified chamber. The sections were washed three times with 1 X PBS for 2 min each, mounted with ProLong Gold Antifade Mountant with DAPI (Life technologies) and imaged with a Nikon Eclipse 50i microscope and an Olympus DP72 camera. To assay cell apoptosis, we performed terminal deoxynucleotidyl transferase dUTP nick end labeling (TUNEL) assay according to the manufacturer's instructions (FragEL DNA Fragmentation Detection Kit, Colorimetric-Klenow Enzyme, Calbiochem). Briefly, after deparaffinization and rehydration, the tissue sections were rinsed in 1 X tris-buffered saline (TBS), permeabilized with Proteinase K and the endogenous peroxidase activity was blocked with 3% hydrogen peroxide for 5 min at room temperature. The sections were then incubated with Klenow Labeling reaction mixture in a humidified chamber at 37 °C for 1.5 hr. For labeling detection, the sections were incubated with peroxidase streptavidin conjugate and subsequently with DAB solution. The sections were counterstained with methyl green solution and imaged with a Nikon Eclipse 50i microscope and an Olympus DP72 camera.

## Second harmonic generation imaging and analysis

Frozen sections were dried at room temperature, washed with 1 X phosphate-buffered saline, and mounted using Aqua-Poly/Mount (Polysciences Inc, cat. 18606) and #1 slip cover (Richard-Allan Scientific, cat. 12460). Using the 40 X oil-objective on a Zeiss LSM 880-Inverted microscope, 40 micron z-stacks were captured from areas centered within the inter-sphenoid synchondrosis region of each sample (n=3–4). In ImageJ, each z-stack was used to create a maximum projection image for anisotropy analysis. Using the polygon tool, 2–4 regions of interest (ROI) were created on each maximum projection image to include areas with positive signal that were not oversaturated. FibrilTool, a plug-in for ImageJ, was used to quantify the anisotropy of collagen in each ROI (*Boudaoud et al., 2014*). For analysis, ROIs were averaged for each sample.

## Statistical analysis

The graphs and statistical analysis were performed in the GraphPad Prism software (version 6.0e, La Jolla California USA). Mouse studies used an N=10 based on power analysis of data from our previous study with *Ddr2*-deficient mice (*Ge et al., 2016*) where we estimate a minimum of 8 animals/group will be required to detect a power of.80 (95% CI, estimated effect size of $\eta$ 2>0.40). All values were presented as mean ± S.D. Unpaired, two-tailed Student's *t* test was used to analyze the difference between the two experimental groups. * p<0.05; ** p<0.01, *** p<0.001, **** p<0.0001; n.s. not significant.

## Acknowledgements

This work was supported by a scholarship from the Ministry of Higher Education and Scientific Research, Libyan Transitional Government (FFM), a scholarship from King Saud University (AB), NIH/

NIDCR grants DE11723, DE029012, DE029465, Department of Defense grant PR190899, research funds from the Department of Periodontics and Oral Medicine, University of Michigan School of Dentistry (to RTF), NIH R01 AR078324 (BL) and the Michigan Musculoskeletal Health Core Center (NIH/NIAMS P30 AR069620).

---

## Additional information

### Funding

| Funder | Grant reference number | Author |
|---|---|---|
| National Institute of Dental and Craniofacial Research | R01DE11723 | Renny T Franceschi |
| National Institute of Dental and Craniofacial Research | R21DE029012 | Renny T Franceschi |
| National Institute of Dental and Craniofacial Research | R01DE029465 | Renny T Franceschi |
| U.S. Department of Defense | PR190899 | Renny T Franceschi |
| National Institute of Arthritis and Musculoskeletal and Skin Diseases | R01AR078324 | Benjamin Levi |
| National Institute of Arthritis and Musculoskeletal and Skin Diseases | P30AR069620 | Renny T Franceschi |
| Ministry of Higher Education and Scientific Research | | Fatma F Mohamed |
| King Saud University | | Abdul-Aziz Binrayes |

The funders had no role in study design, data collection and interpretation, or the decision to submit the work for publication.

### Author contributions

Fatma F Mohamed, Conceptualization, Formal analysis, Investigation, Methodology, Writing – original draft, Writing – review and editing; Chunxi Ge, Investigation, Supervision, Conceptualization, Formal analysis, Writing – original draft; Shawn A Hallett, Abdul-Aziz Binrayes, Benjamin Levi, Conceptualization, Writing – original draft; Alec C Bancroft, Conceptualization; Randy T Cowling, Resources, Writing – review and editing; Noriaki Ono, Resources, Formal analysis, Writing – original draft; Barry Greenberg, Resources, Writing – original draft; Vesa M Kaartinen, Supervision, Writing – original draft; Renny T Franceschi, Conceptualization, Formal analysis, Funding acquisition, Project administration, Supervision, Validation, Writing – original draft, Writing – review and editing

### Author ORCIDs

Shawn A Hallett  http://orcid.org/0000-0003-1472-7502
Noriaki Ono  http://orcid.org/0000-0002-3771-8230
Renny T Franceschi  https://orcid.org/0000-0003-1405-2541

### Ethics

This study was performed in strict compliance with the Guidelines for the Care and Use of Animals for Scientific Research. All of the animals were handled according to approved institutional animal care and use committee (IACUC) protocols (PRO9305, PRO10975) of the University of Michigan.

### Decision letter and Author response

Decision letter https://doi.org/10.7554/eLife.77257.sa1
Author response https://doi.org/10.7554/eLife.77257.sa2

---

# Additional files

## Supplementary files
• Transparent reporting form

## Data availability
All data generated or analysed during this study are included in the manuscript and source data files.

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
