## [Editor Report]

This fundamental work substantially advances our understanding of the development of the craniofacial skeleton requires interactions between progenitor cells and the collagen-rich extracellular matrix (ECM). The evidence supporting the conclusions is compelling, with rigorous biochemical assays, state-of-the-art localization and lineage tracing studies, and phenotype and genetic analysis of Ddr2-deficient mice. The work will be of broad interest to cell biologists and developmental biologists.

---

## [Decision Letter]

**Decision letter after peer review:**

Thank you for submitting your article "Control of Craniofacial Development by the Collagen Receptor, Discoidin Domain Receptor 2" for consideration by *eLife*. Your article has been reviewed by 3 peer reviewers, and the evaluation has been overseen by a Reviewing Editor and Mone Zaidi as the Senior Editor. The manuscript is of interest to *eLife*. The reviewers have discussed their reviews with one another, and the Reviewing Editor has drafted this to help you prepare a revised submission. The following individual involved in the review of your submission has agreed to reveal their identity: Aaron T Blanchard (Reviewer #2).

Essential revisions:

We would like to suggest that the authors revise the manuscript and submit a revised manuscript to our journal. Following are the essential revision requirements:

– Provide major new data to show previously unknown cellular and molecular mechanisms underlying Ddr2 signaling in Gli1+ skeletal progenitor cells and chondrocytes. These mechanisms should explain chondrocyte proliferation and polarity as well as osteoblast differentiation (see reviewer 1's comments).

– Provide more rigorous quantification and more clear histological evidence.

– Add the Discussion section would ideally include a broader literature survey that considers work on the molecular biophysics of DDR2.

– More clarifications in term of methods might be required and needs some extra focus on the writing points (see reviewer 3's comments.)

*Reviewer #1 (Recommendations for the authors):*

In order for this manuscript to be considered for publication in *eLife*, the authors must provide major new data to show previously unknown cellular and molecular mechanisms underlying Ddr2 signaling in Gli1+ skeletal progenitor cells and chondrocytes. These mechanisms should explain chondrocyte proliferation and polarity as well as osteoblast differentiation.

*Reviewer #3 (Recommendations for the authors):*

Title

Line 2, better to modify the title to become (Investigate role of Collagen Receptor, Discoidin Domain 3 Receptor 2 to control Craniofacial Development).

Abstract

Line 47, Capitalize the first letter of (DDR2) receptor name.

Preferable to refer briefly to the main results of the study at the end of the abstract.

Introduction

Line 95, this process is harmonized with…

Line 118. Moreover, DDR2 loss of function is the main reason behind the autosomal recessive.

Line 123, you don't have to mention preposition (and) between all abnormalities, you only need it between the last two abnormalities.

Line 132, We show that DDR2 is …

Line 137, Notably, functions of DDR2 involve activity in different cell populations.

Results

Line 140, Anterior-posterior skull growth and frontal suture/bone formation impairment in Ddr2-deficient mice.

Line 186, To begin understand.

Line 190, Dd2?

Line 191-192, using Cleaved Cspase3 might give you the best results for apoptosis measurement.

Line 217, In order to investigate the related of the cellular functions …

Line 244, Indication profound function of Ddr2 in skeletal progenitor cells …

Line 247, titles of results paragraph should more precise, and it doesn't have to be as long as this.

Line 251, we would determine whether Ddr2 is in skeletal progenitor cells whose 252 progeny can form the major cranial bone cell types …

Line 285, to answer this question …

Line 303, Ddr2 roles in Gli1+ skeletal progenitors to control craniofacial morphogenesis…

Line 305, Based on aforementioned results, we conclude …

Line 309, you may need a reference at end of the sentence …

Line 338, same note above for GLI1…

Discussion

Line 529, seen in Ddr2-deficient mice …

Line 532 to 536, long sentence!

Line 540, Moreover, disruption of GM130 in mitosis …

Line 542, although these mechanisms required more clarification…

Line 573, skeleton, and advance our understanding.

Materials and methods

Ethical approval is required in a separate paragraph.

---

## [Author Response]

Essential revisions:We would like to suggest that the authors revise the manuscript and submit a revised manuscript to our journal. Following are the essential revision requirements:– Provide major new data to show previously unknown cellular and molecular mechanisms underlying Ddr2 signaling in Gli1+ skeletal progenitor cells and chondrocytes. These mechanisms should explain chondrocyte proliferation and polarity as well as osteoblast differentiation (see reviewer 1's comments).

Please see response to Reviewer 1. Important new data is now included showing that organization of the type II collagen matrix in synchondroses of Ddr2-deficient mice is disrupted resulting in decreased anisotropy of collage fibrils as measured by second harmonic generation microscopy. Studies are also cited showing that the phenotypes of mice harboring Col2a1 mutations that disrupt ECM organization are similar to those seen in Ddr2-deficient mice including loss of chondrocyte polarity.

– More clarifications in term of methods might be required and needs some extra focus on the writing points (see reviewer 3's comments.)

Please see response to Reviewer 3.

Reviewer #1 (Recommendations for the authors):In order for this manuscript to be considered for publication in eLife, the authors must provide major new data to show previously unknown cellular and molecular mechanisms underlying Ddr2 signaling in Gli1+ skeletal progenitor cells and chondrocytes. These mechanisms should explain chondrocyte proliferation and polarity as well as osteoblast differentiation.

Thank you for this comment. Very little is known about Ddr2 signaling in bone or any other tissue and extensive additional studies outside the scope of the present work would be necessary to provide a definitive mechanism to explain our results. However, new data included in the manuscript show that Ddr2 deficiency leads to disruption of collagen organization and orientation as measured by second harmonic generation (SHG) (Figure 3—figure supplement 1). Specifically, collagen orientation as reflected by SHG anisotropy measurements was disrupted in Ddr2-deficient synchondroses. This result complements data showing that the distribution of type II collagen as measured by immunofluorescence changes with Ddr2 deficiency such that no collagen is seen in the interterritorial matrix between chondrocyte bundles (Figure 3a). Ln 207-218

This loss of collagen organization provides a potential mechanism to explain the disruption of chondrocyte polarity and altered localization of hypertrophic cells in synchondroses. In further support of this concept, other recently published studies described in the Discussion have shown that Ddr2 deficiency is associated with disruption of collagen fibril orientation in other experimental systems such as in CAF cells surrounding breast tumors as well as at sites of heterotopic ossification and that these abnormalities are associated with defective integrin signaling. Additional studies beyond the scope of the present communication will be required to determine if these matrix changes can explain the observed phenotypes. However, we believe this proposed mechanism is the most likely explanation for DDR2 effects based on current data.

The Discussion now contains a section on the relationships between DDR2, ECM orientation and cell behavior that summarizes current knowledge in this area. Ln 555-605

Reviewer #3 (Recommendations for the authors):TitleLine 2, better to modify the title to become (Investigate role of Collagen Receptor, Discoidin Domain 3 Receptor 2 to control Craniofacial Development).

Thank you for the suggestion. However, we think our current title adequately describes our study.

AbstractLine 47, Capitalize the first letter of (DDR2) receptor name.

We have used the standard convention for mouse genes and proteins (*Ddr2* gene, DDR2 protein).

Preferable to refer briefly to the main results of the study at the end of the abstract.

Abstract has been modified to summarize overall results of this study.

IntroductionLine 95, this process is harmonized with…

Thank you for the suggestion. However, our original wording captures our intended meaning. “This process is coordinated with growth of the cranial base mediated by endochondral ossification centers called synchondroses.” Ln 96-97

Line 118. Moreover, DDR2 loss of function is the main reason behind the autosomal recessive.

Change made. Thank you for the suggestion. Ln 121-123

Line 123, you don't have to mention preposition (and) between all abnormalities, you only need it between the last two abnormalities.

Change made. Thank you for the suggestion. Ln 126-128

Line 132, We show that DDR2 is …

Change made. Thank you for the suggestion. Ln 136-139

Line 137, Notably, functions of DDR2 involve activity in different cell populations.

Change made. Thank you for the suggestion. Ln 141-143

ResultsLine 140, Anterior-posterior skull growth and frontal suture/bone formation impairment in Ddr2-deficient mice.

Change made. Thank you for the suggestion. Ln 145

Line 186, To begin understand.

Change made. Thank you for the suggestion. Ln 190

Line 190, Dd2?

Typo corrected. Thank you. ln 194

Line 191-192, using Cleaved Cspase3 might give you the best results for apoptosis measurement.

Thank you for the suggestion. As requested, we repeated apoptosis analysis using immunostaining for cleaved caspase 3. In agreement with results of TUNEL assay, no significant changes were seen between wild type and Ddr2-deficient mice. Figure 2-Supplemental Figure 1 and Ln 197-198

Line 217, In order to investigate the related of the cellular functions …

Thank you for the suggestion. However, we believe the sentence we use more accurately conveys the purpose of these studies.

Ln 233 “To begin relating the cellular functions of DDR2 to the observed craniofacial phenotype of *Ddr2* deficient mice, we first examined the temporal and spatial distribution of *Ddr2*-expressing cells using a *Ddr2*-lacZ knock-in mouse model.”

Line 244, Indication profound function of Ddr2 in skeletal progenitor cells …

Thank you for the suggestion. However, we believe the sentence we use more accurately conveys our intended meaning.

Ln 259-261 “This suggests that *Ddr2* has functions in skeletal progenitor cells which contribute to development of the craniofacial skeleton.”

Line 247, titles of results paragraph should more precise, and it doesn't have to be as long as this.

Thank you for the suggestion. Titles have been shortened where possible. Ln 171, 263

Line 251, we would determine whether Ddr2 is in skeletal progenitor cells whose 252 progeny can form the major cranial bone cell types …

Thank you for the suggestion. Sentence has been modified Ln 266-267

“Since *Ddr2* is also expressed in sutures, we determined whether DDR2 is in skeletal progenitor cells whose progeny can form the major cranial bone cell types.”

Line 285, to answer this question …

Thank you for the suggestion. Change made as suggested. Ln 267-269

Line 303, Ddr2 roles in Gli1+ skeletal progenitors to control craniofacial morphogenesis …

Thank you for the suggestion. However, we believe the sentence we use more accurately conveys our intended meaning. Ln 322

Line 305, Based on aforementioned results, we conclude …

Thank you for the suggestion. Change made as suggested. Ln 324

Line 309, you may need a reference at end of the sentence …

This sentence refers to studies in this manuscript, not the literature. To clarify, the sentence has been modified to:

“Furthermore, our localization and lineage tracing studies suggest preferential expression of *Ddr2* in suture-associated skeletal progenitors and resting/proliferating zone chondrocytes in cranial base synchondroses.” Ln 326-329

Line 338, same note above for GLI1…

Again this refers to the current manuscript. Ln 359

DiscussionLine 529, seen in Ddr2-deficient mice …

Sentence removed from rewrite.

Line 532 to 536, long sentence!

Sentence removed from rewrite.

Line 540, Moreover, disruption of GM130 in mitosis …

Change made. Thank you. ln 570

Line 542, although these mechanisms required more clarification…

Change made. Thank you. ln 571-572

Line 573, skeleton, and advance our understanding.

Change made. Thank you. ln 608

Materials and methodsEthical approval is required in a separate paragraph.

Change made. Thank you. ln 630